# SSCBench: Monocular 3D Semantic Scene Completion Benchmark in Street Views

## Abstract

Monocular scene understanding is a foundational component of autonomous systems. Within the spectrum of monocular perception topics, one crucial and useful task for holistic 3D scene understanding is semantic scene completion (SSC), which jointly completes semantic information and geometric details from RGB input. However, progress in SSC, particularly in large-scale street views, is hindered by the scarcity of high-quality datasets. To address this issue, we introduce SSCBench, a comprehensive benchmark that integrates scenes from widely used automotive datasets (e.g., KITTI-360, nuScenes, and Waymo). SSCBench follows an established setup and format in the community, facilitating the easy exploration of SSC methods in various street views. We benchmark models using monocular, trinocular, and point cloud input to assess the performance gap resulting from sensor coverage and modality. Moreover, we have unified semantic labels across diverse datasets to simplify cross-domain generalization testing. We commit to including more datasets and SSC models to drive further advancements in this field.

## 1 Introduction

Understanding 3D scenes from a single RGB image is crucial and meaningful in vision and robotics, with monocular perception tasks like object detection (Wang et al., 2022), tracking (Hu et al., 2022), and depth estimation (Yuan et al., 2022) garnering significant attention. The emerging field of 3D semantic scene completion (SSC) (Roldao et al., 2022) seeks to jointly infer complete 3D semantics and geometry from a sparse and partial observation (*e.g.*, an RGB image). The resulting volumetric representation seamlessly integrates occupancy and semantic information, facilitating robotic scene understanding and planning capabilities in street views.

One critical challenge in SSC is to generate accurate ground truth labels, especially in street views. Given the current limitations of 3D sensing technology, achieving a perfectly comprehensive 3D representation is impossible. The pioneering SemanticKITTI benchmark (Behley et al., 2019) proposes to leverage the temporal information through the aggregation of different LiDAR sweeps, which can effectively reveal previously occluded 3D surfaces. Meanwhile, it excludes 3D voxels not observed from all viewpoints during driving. Consequently, SemanticKITTI provides relatively comprehensive and accurate ground truth labels for SSC tasks.

While SemanticKITTI is a valuable resource for learning sparse-to-dense mapping, its limited scale and diversity impede the development of more powerful and generalizable SSC models. Another significant limitation of SemanticKITTI is the omission of dynamic objects during ground truth generation, resulting in inaccurate labels. Hence, there is an urgent need for a large-scale SSC dataset with reliable ground truth to advance learning-based scene understanding in street views.

To this end, we introduce SSCBench, a large-scale benchmark comprising diverse street views sourced from well-established automotive datasets, including KITTI-360 (Liao et al., 2022), nuScenes (Caesar et al., 2020), and Waymo (Sun et al., 2020), as illustrated in Fig. 1. To enhance label accuracy, we utilize the 3D bounding box labels provided in these datasets to synchronize measurements of dynamic objects. Our key features include **(a)** *accessibility*: we provide datasets in a format compatible with SemanticKITTI, facilitating seamless usage within the community; **(b)** *large scale*: we offer an extensive dataset with ∼8 times more frames than SemanticKITTI, encompassing diverse geographic locations across six cities; **(c)** *comprehensiveness*: we mainly focus on SSC methods with *monocular input*. Additionally, we utilize *trinocular input* to compare the single-view and panoramic-view methods and use *point cloud input* to show the gap between camera-based and

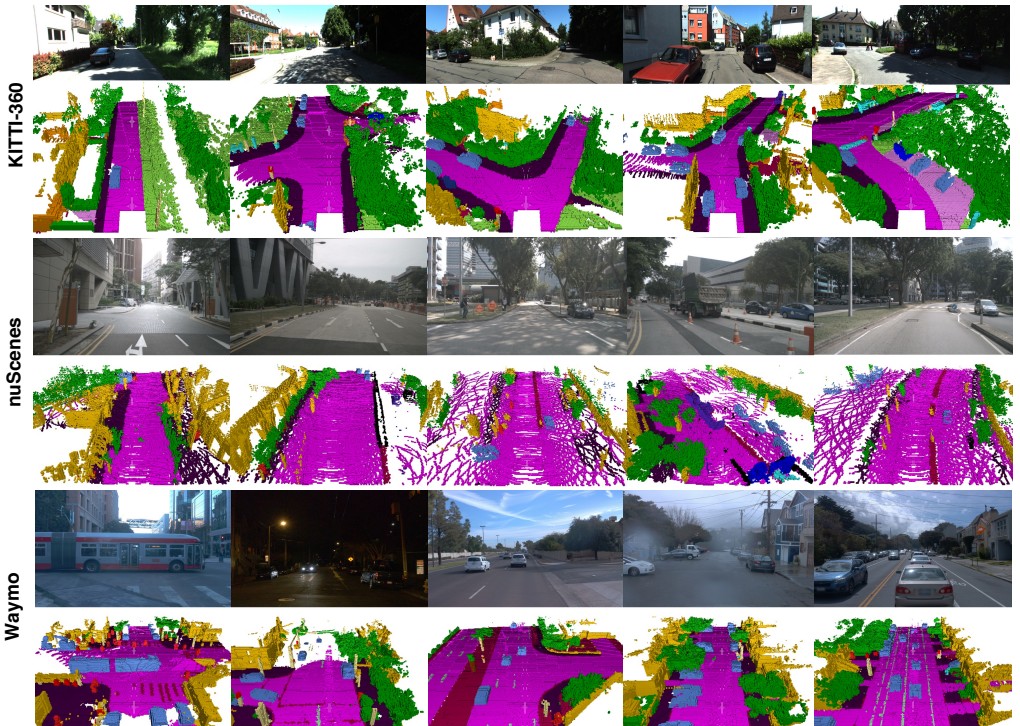

Figure 1: **Visualizations of SSCBench** derived from KITTI-360 (Liao et al., 2022), nuScenes (Caesar et al., 2020), and Waymo (Sun et al., 2020). We showcase accurate SSC ground truth in a variety of street views. We commit to incorporating novel datasets for further development.

LiDAR-based methods. Furthermore, we have unified semantic labels across different datasets in SSCBench, facilitating cross-domain generalization experiments. We plan to continually incorporate novel automotive datasets and SSC algorithms to drive further advancements in the field.

## 2 RELATED WORKS

**Monocular Perception and 3D Semantic Scene Completion.** The simplicity, efficiency, afford-ability, and accessibility of monocular cameras have made monocular perception a focal point of attention in the vision and robotics community. This has resulted in extensive research into various tasks, including depth estimation (Dong et al., 2022), 3D object detection and tracking (Enzweiler & Gavrila, 2008), as well as localization and mapping (Younes et al., 2017). Song et al. (2017) introduce the concept of monocular 3D semantic scene completion (SSC), which seeks to reconstruct and complete the semantics as well as geometry within a 3D volume from a single depth image. However, they only consider the bounded indoor scenarios due to the lack of outdoor datasets. Behley et al. (2019) build the first outdoor dataset based on KITTI (Geiger et al., 2012) for 3D semantic scene completion in street views. Existing approaches usually depend on 3D inputs, such as LiDAR point clouds (Roldao et al., 2020; Cheng et al., 2021a; Rist et al., 2021; Yan et al., 2021), while recent monocular vision-based solutions also emerge (Cao & de Charette, 2022; Li et al., 2023). However, the development of outdoor SSC is hindered by the lack of datasets, with SemanticKITTI (Behley et al., 2019) being the only dataset supporting SSC in street views. Building diverse datasets is imperative to unlock the full potential of SSC for autonomous systems.

**Point Cloud Segmentation in Street Views.** 3D LiDAR segmentation aims to assign point-wise semantic labels for point clouds, including a range of specific tasks, like LiDAR semantic (Hu et al., 2020; Qiu et al., 2021; Milioto et al., 2019; Xu et al., 2020; Cortinhal et al., 2020), panoptic (Zhou et al., 2021; Hong et al., 2021; Behley et al., 2021), and 4D panoptic segmentation (Kreuzberg et al., 2022; Aygun et al., 2021). In this field, point-based methods, stemming from PointNet++ (Qi et al., 2017), perform well on small synthetic point cloud (Yi et al., 2016) rather than sparse LiDAR point cloud, with sampling and gathering disordered neighbors. Voxel-based approaches (Zhu et al., 2021; Tang et al., 2020; Li et al., 2022a; Cheng et al., 2021b) process point clouds by initially partitioning 3D space into voxels through Cartesian coordinates. Note that 3D LiDAR segmentation aims to

Table 1: **Overview of widely-used autonomous driving datasets with multimodal sensors.** C denotes camera and L denotes LiDAR. Most datasets provide bounding annotations for 3D detection, yet only a few of them provide semantic labels for the LiDAR point cloud due to the high cost. Note that ApolloScape (Huang et al., 2019) only provides 3D semantic labels for the static environments.

| Datasets | Year | Sensors | Annotations | # Fr. with Pts Ann. | Sequential |
|---|---|---|---|---|---|
| KITTI Geiger et al. (2012) | CVPR 2012 | C&L | 3D Bbox | N.A. | ✗ |
| SemanticKITTI Behley et al. (2019) | ICCV 2019 | C&L | 3D Pts. | 20K | ✓ |
| nuScenes Caesar et al. (2020) | CVPR 2019 | C&L | 3D Bbox | N.A. | ✓ |
| Panoptic nuScenes Fong et al. (2022) | RA-L 2022 | C&L | 3D Pts. | 40K | ✓ |
| Waymo Sun et al. (2020) | CVPR 2020 | C&L | 3D Bbox&Pts. | 230K | ✓ |
| KITTI-360 Liao et al. (2022) | T-PAMI 2022 | C&L | 3D Bbox&Pts. | 100K | ✓ |
| ApolloScape Huang et al. (2019) | T-PAMI 2019 | C&L | 3D Bbox&Pts. | N.A. | ✓ |
| Argoverse Chang et al. (2019) | CVPR 2019 | C&L | 3D Bbox | N.A. | ✓ |
| ONCE Mao et al. | NeurIPS 2021 | C&L | 3D Bbox | N.A. | ✓ |
| Lyft Level 5 Kesten et al. (2019) | 2019 | C&L | 3D Bbox | N.A. | ✓ |
| A*3D Pham et al. (2020) | ICRA 2020 | C&L | 3D Bbox | N.A. | ✓ |
| A2D2 Geyer et al. (2020) | 2020 | C&L | 3D Bbox | N.A. | ✗ |

classify and understand the scenes based on raw LiDAR scans, while 3D semantic scene completion includes the completion of occluded areas, with input of camera or LiDAR.

**Autonomous Driving Dataset and Benchmark.** Autonomous driving research thrives on high-quality datasets, which serve as the lifeblood for training and evaluating perception (Caesar et al., 2020), prediction (Ettinger et al., 2021), and planning algorithms (Hu et al., 2023). In 2012, the pioneering KITTI dataset sparked a revolution in autonomous driving research, unlocking a multitude of tasks including object detection, tracking, mapping, and optical/depth estimation (Geiger et al., 2012; 2013; Fritsch et al., 2013; Menze & Geiger, 2015; Chen et al., 2023). Since then, the research community has embraced the challenge, giving rise to a wealth of datasets. These datasets push the boundaries of autonomous driving research by addressing challenges posed by multimodal fusion (Caesar et al., 2020), multi-tasking learning (Huang et al., 2018; Liao et al., 2022), adverse weather (Pitropov et al., 2021), collaborative driving (Agarwal et al., 2020; Li et al., 2022b; Xu et al., 2023), repeated driving (Diaz-Ruiz et al., 2022), and dense traffic scenarios (Pham et al., 2020; Xiao et al., 2021), *etc*. There are several impactful and widely-used driving datasets such as KITTI-360 (Liao et al., 2022), nuScenes (Caesar et al., 2020), and Waymo (Sun et al., 2020). They provide LiDAR and camera recordings as well as point cloud semantics and bounding annotations, as summarized in Tab. 1. Therefore, we can create accurate ground truth labels for SSC by aggregating multiple semantic point clouds and leveraging the 3D boxes to handle dynamic objects.

**Occ3D and OpenOccupancy.** We compare SSCBench with the concurrent relevant work Occ3D (Tian et al., 2023). The differences lie in: *(a) setup*: Occ3D uses surrounding-view images as input, and only considers the reconstruction of 3D voxels visible to the camera. SSCBench considers a more challenging yet meaningful setup (also a well-established one): how to reconstruct and complete 3D semantics in both visible and occluded areas only with monocular visual input. *This task requires reasoning about temporal information and 3D geometric relationships to get rid of the limited field of view*; *(b) scale*: SSCBench provides more datasets than Occ3D and plans to add more due to the abundance of monocular driving recordings; *(c) accessibility*: we inherit the widely-used setup from the pioneer KITTI, thus making SSCBench more accessible to the community; *(d) comprehensiveness*: we benchmark SSC methods with monocular, trinocular, and point cloud input and provide unified labels for cross-domain generalization tests. Another relevant benchmark, OpenOccupancy (Wang et al., 2023), exhibits similar differences, notably its exclusive use of the nuScenes dataset (Caesar et al., 2020), which results in a limitation of diversity.

## 3 DATASET CURATION

### 3.1 REVISIT OF SEMANTICKITTI

SemanticKITTI (Behley et al., 2019) extends the odometry dataset of the KITTI vision benchmark (Geiger et al., 2012) by providing point-wise semantic annotations for 22 driving sequences in

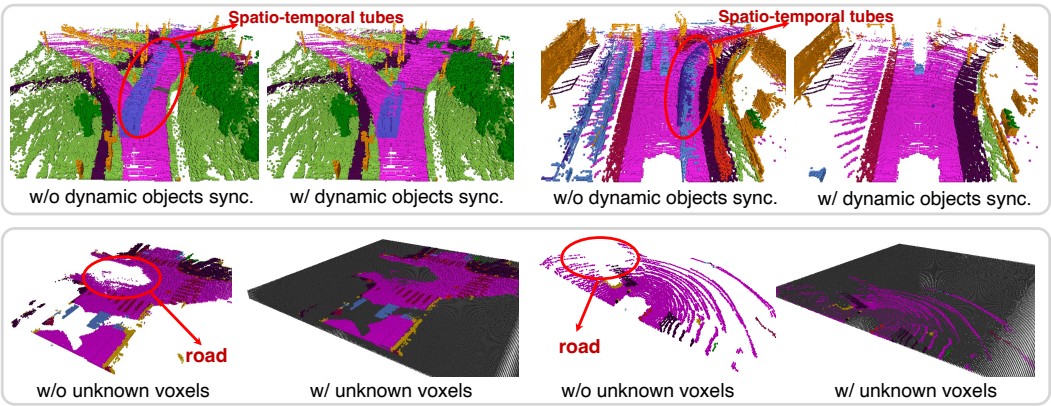

Figure 2: **Top row: dynamic objects synchronization.** Two examples on nuScenes (Caesar et al., 2020) are shown. Spatio-temporal tubes are introduced without handling dynamic objects, damaging the accuracy of labels. **Bottom row: unknown voxels exclusion.** Voxels are marked as unknown (denoted by grey color) when they are occluded by other objects or remain unprobed by the LiDAR.

Karlsruhe, Germany. SemanticKITTI not only supports 3D semantic segmentation but also serves as the first outdoor SSC benchmark. Similar to indoor SSC (Song et al., 2017), SemanticKITTI uses voxelized 3D representation widely employed in robotics such as occupancy grid mapping (Thrun, 2002). SemanticKITTI generates ground truth labels through voxelization of a dense semantic point cloud given by rigid registration of multiple LiDAR scans.

SemanticKITTI has two limitations. First, rigid registration with sensor poses can only handle measurements for static environments, resulting in traces produced by dynamic objects such as moving cars as shown in Fig. 2, which can confuse the 3D representation learning (Rist et al., 2021). Secondly, it is constrained by the limited scale and lack of diverse geographical coverage. The data collection is confined to a single city, resulting in training, validation, and test sets composed of 3,834, 815, and 3,992 frames, respectively, amounting to a total of 8,641 frames. However, this falls short of the large-scale benchmark necessary for comprehensive evaluation and generalization in the field.

### 3.2 SSCBENCH

We aim to establish a large-scale SSC benchmark in street views that facilitates the training of robust and generalizable SSC models. To achieve this, we harness well-established and widely-used datasets and integrate them into a unified setup and format. Overall, our SSCBench, consisting of three subsets, includes 38,562 frames for training, 15,798 frames for validation, and 12,553 frames for testing respectively, amounting totally to 66,913 frames ($\sim$67K), which greatly exceeds the scale of SemanticKITTI mentioned above by $\sim$7.7 times. In the following, we introduce three carefully designed datasets, all based on existing data sources, that collectively contribute to our SSCBench.

**SSCBench-KITTI-360.** KITTI-360 (Liao et al., 2022) represents a significant advancement in autonomous driving research, building upon the renowned KITTI dataset (Geiger et al., 2012). It introduces an enriched data collection framework with diverse sensor modalities as well as panoramic viewpoints (a perspective stereo camera plus a pair of fisheye cameras) and provides comprehensive annotations including consistent 2D and 3D semantic instance labels as well as 3D bounding primitives. The dense and coherent labels not only support established tasks such as segmentation and detection but also enable novel applications like semantic SLAM (Bowman et al., 2017) and novel view synthesis (Zhang et al., 2023a). While KITTI-360 includes point cloud-based semantic scene completion, the prevalent methodology for SSC remains centered around voxelized representations (Roldao et al., 2022), which exhibit broader applicability in robotics.

*Remark.* KITTI-360 covers a driving distance of 73.7km, comprising 300K images and 80K laser scans. While adhering to KITTI's forward-facing camera setup, it offers greater geographical diversity and demonstrates minimal trajectory overlap with KITTI. Leveraging the open-source training and validation set, we build SSCBench-KITTI-360 consisting of 9 long sequences. To reduce redundancy, we sample every 5 frames following the SemanticKITTI SSC benchmark. The training set includes

8,487 frames from scenes 00, 02-05, 07, and 10, while the validation set comprises 1,812 frames from scene 06. The testing set comprises 2,566 frames from scene 09. In total, the dataset contains 12,865 (∼13K) frames, surpassing the scale of SemanticKITTI by ∼1.5 times.

**SSCBench-nuScenes.** Unlike KITTI's forward-facing camera setup, nuScenes (Caesar et al., 2020) captures a complete 360-degree view around the ego vehicle. It provides a diverse range of multimodal sensory data, including camera images, LiDAR point clouds, and radar data, gathered in Boston and Singapore. nuScenes offers meticulous annotations for complex urban driving scenarios, including diverse weather conditions, construction zones, and varying illumination. Later on, panoptic nuScenes (Fong et al., 2022) extends the original nuScenes dataset with semantic and instance labels. With comprehensive metrics and evaluation protocols, nuScenes is widely employed in autonomous driving research (Gu et al., 2023; Hu et al., 2023; Li et al., 2021; Huang et al., 2023).

*Remark.* The nuScenes dataset consists of 1K 20-second scenes with labels provided only for the training and validation set, totaling 850 scenes. From the available 850 scenes, we allocate 500 scenes for training, 200 scenes for validation, and 150 scenes for testing. This distribution results in 20,064 frames for training, 8,050 frames for validation, and 5,949 frames for testing, totaling 34,078 frames (∼34K). This scale is approximately four times that of SemanticKITTI. As nuScenes only provides annotations for keyframes at a frequency of 2Hz, there is no downsampling in SSCBench-nuScenes.

**SSCBench-Waymo.** The Waymo dataset (Sun et al., 2020), collected from various locations in the US, offers a large-scale collection of multimodal sensor recordings. Waymo provides 5 cameras with a combined horizontal field of view of ∼230 degrees, slightly smaller than nuScenes. The data is captured in diverse conditions across multiple cities, including San Francisco, Phoenix, and Mountain View, ensuring broad geographical coverage within each city. It consists of 1000 scenes for training and validation, as well as 150 scenes for testing, with each scene spanning 20 seconds.

*Remark.* To construct SSCBench-Waymo, we utilize the open-source training and validation scenes and redistribute them into sets of 500, 298, and 202 scenes for training, validation, and testing, respectively. In order to reduce redundancy and training time for our benchmark, we downsample the original data by a factor of 10. This downsampling results in a training set of 10,011 frames, a validation set of 5,936 frames, and a test set of 4,038 frames, totaling 19,985 frames (∼20K).

### 3.3 CONSTRUCTION PIPELINE

**Prerequisites.** To establish SSCBench, a driving dataset with multimodal recordings is required for LiDAR-based or camera-based SSC. The dataset should include sequentially collected 3D LiDAR point clouds with accurate sensor poses for geometry completion, per-point semantic annotations for semantic scene understanding, and 3D bounding annotations to handle dynamic instances.

**Aggregation of Point Clouds.** To generate a complete representation, our approach involves superimposing an extensive set of laser scans within a defined region in front of the vehicle. In short sequences like nuScenes and Waymo, we utilize future scans with measurements from the corresponding region to create a dense semantic point cloud. In long sequences like KITTI-360, which feature multiple loop closures, we incorporate all spatial neighboring point clouds in addition to the temporal neighborhood. Accurate sensor poses, provided by advanced SLAM systems (Bailey & Durrant-Whyte, 2006), greatly facilitate the aggregation of point clouds for the static environment. As for dynamic objects, we avoid the spatial-temporal tubes by synchronization. We utilize the instance label to transform dynamic objects to their spatial alignment within the current frame. As shown in Fig. 2, the spatial-temporal tubes are removed and the objects have a denser shape.

**Voxelization of Aggregated Point Clouds.** Voxelization is to discretize a continuous 3D space into a regular grid structure composed of volumetric elements called voxels, enabling the conversion of unstructured data into a structured format that can be efficiently processed by convolutional neural networks (CNNs) or vision transformers (ViTs). Voxelization introduces a trade-off between spatial resolution and memory consumption and offers a flexible and scalable representation for 3D perception (Maturana & Scherer, 2015; Zhou & Tuzel, 2018; Li et al., 2023). For easy integration, SSCBench adheres to SemanticKITTI's setup, with a volume extending 51.2m ahead, 25.6m on each side, and 6.4m in height. Voxel resolution is 0.2m, resulting in a $256 \times 256 \times 32$ voxel volume. Labels for each voxel are determined by majority voting among labeled points within it, while empty voxels are marked accordingly if no points are present.

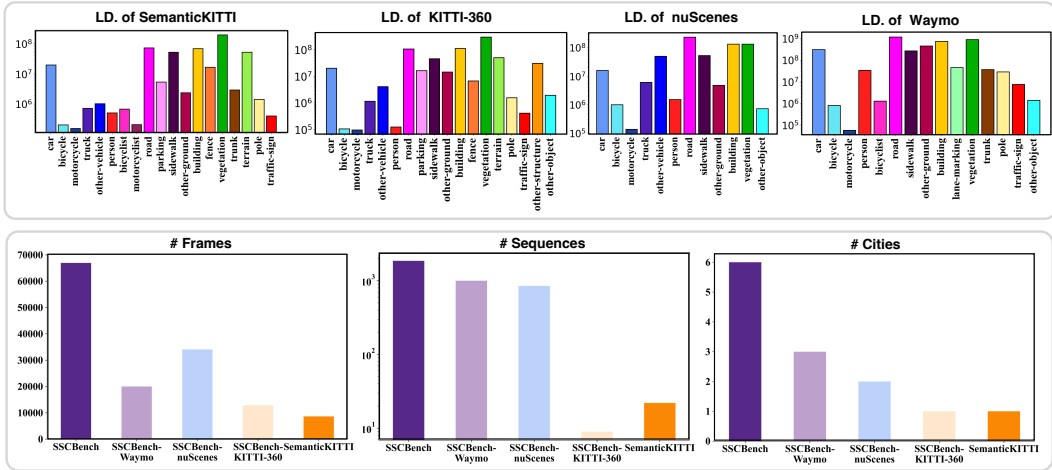

Figure 3: **Statistical analysis.** Top row: Label Distribution (LD.) of different datasets. Bottom row: scale comparisons between SSCBench and SemanticKITTI.

**Exclusion of Unknown Voxels.** Capturing complete 3D outdoor dynamic scenes is nearly impossible without ubiquitous scene sensing. While it is possible to utilize spatial or prior knowledge-based inference, our intention is to ensure the fidelity of ground truth by minimizing errors originating from these steps. Hence, we only consider *visible and probed* voxels from all viewpoints during training and evaluation. Specifically, we first employ ray tracing from different perspectives to identify and remove occluded voxels within objects or behind walls. Furthermore, in datasets with sparse sensing, where numerous voxels remain unprobed, we remove these unknown voxels during training and evaluation to enhance the reliability of the ground truth labels as shown in Fig. 2.

### 3.4 DATASET STATISTICS

We show dataset statistics in Fig. 3. The dataset label distribution reveals noticeable domain gaps among various cities. Specifically, KITTI-360 and SemanticKITTI exhibit similar label distributions due to being captured within the same city in Germany. However, nuScenes and Waymo collected in the US and Singapore demonstrate distinct label distributions. Furthermore, SSCBench stands out for its larger scale in comparison to SemanticKITTI. For instance, SSCBench comprises 7.7 times more frames than SemanticKITTI, and its collection of driving sequences is also more diverse.

## 4 EXPERIMENTAL SETUP

**Benchmark Methods.** Our benchmark employs four camera-based methods, *i.e.*, MonoScene (Cao & de Charette, 2022), VoxFormer (Li et al., 2023), TPVFormer (Huang et al., 2023), and Occ-Former (Zhang et al., 2023b), as well as two LiDAR-based methods, *i.e.*, SSCNet (Song et al., 2017) and LMSCNet (Roldao et al., 2020), due to their widespread adoption and cutting-edge performance. Using their public codebases, we run them under default settings but appropriately adjust data loaders and batch sizes to align with our SSCBench. We separately train, validate, and test these methods on SSCBench subsets, as reported in Sec. 5. Furthermore, we provide a unified benchmark for evaluating cross-domain generalizability in Sec. 6, where models are trained on one subset and tested on others, *e.g.*, trained on SSCBench-KITTI-360 and tested on SSCBench-Waymo.

**Evaluation Metrics.** We adopt the intersection over union (IoU) for evaluating geometry completion and the mean IoU (mIoU) of each class for evaluating semantic segmentation, following SemanticKITTI. We also report the ratio of different classes in the dataset to better understand the relationship between IoU and mIoU. We report the performances within the volume extending 51.2m ahead, 25.6m on each side, and 6.4m in height, and the voxel resolution is 0.2m. The design of this front-view evaluation emphasizes the area directly in the vehicle's anticipated forward trajectory. Experimental results across different ranges, *i.e.*, short-range areas ($12.8 \times 12.8 \times 6.4 m^3$) and medium-range areas ($25.6 \times 25.6 \times 6.4 m^3$), are reported in the Appendix. This range-based evaluation provides valuable insights into performance concerning spatial distance.

Table 2: **Separate benchmarking results on three SSCBench subsets.** We benchmark models with *monocular images* and *point cloud* inputs. The default evaluation range is $51.2{\times}51.2{\times}6.4\text{m}^3$. Due to the label differences amongst the three subsets, missing labels are replaced with "-". The top three performances on each dataset are marked by red, green, and blue respectively.

| Dataset | Method | Input | IoU | mIoU | car | bicycle | motorcycle | truck | other-veh. | person | road | parking | sidewalk | other-grnd | building | fence | vegetation | terrain | pole | traf-sign | other-struct. | other-object | bicyclist | lane-marking | trunk |
|---|---|---|---|---|---|---|---|---|---|---|---|---|---|---|---|---|---|---|---|---|---|---|---|---|---|
| SSCBench-KITTI-360 | LMSCNet | L | 47.53 | 13.65 | 20.91 | 0 | 0 | 0.26 | 0 | 0 | 62.95 | 13.51 | 33.51 | 0.2 | 43.67 | 0.33 | 40.01 | 26.80 | 0 | 0 | 3.63 | 0.0003 | - | - | - | - |
| | SSCNet | L | 53.58 | 16.95 | 31.95 | 0 | 0.17 | 10.29 | 0.58 | 0.07 | 65.7 | 17.33 | 41.24 | 3.22 | 44.41 | 6.77 | 43.72 | 28.87 | 0.78 | 0.75 | 8.60 | 0.67 | - | - | - | - |
| | MonoScene | C | 37.87 | 12.31 | 19.34 | 0.43 | 0.58 | 8.02 | 2.03 | 0.86 | 48.35 | 11.38 | 28.13 | 3.22 | 32.89 | 3.53 | 26.15 | 16.75 | 6.92 | 5.67 | 4.20 | 3.09 | - | - | - | - |
| | Voxformer | C | 38.76 | 11.91 | 17.84 | 1.16 | 0.89 | 4.56 | 2.06 | 1.63 | 47.01 | 9.67 | 27.21 | 2.89 | 31.18 | 4.97 | 28.99 | 14.69 | 6.51 | 6.92 | 3.79 | 2.43 | - | - | - | - |
| | TPVFormer | C | 40.22 | 13.64 | 21.56 | 1.09 | 1.37 | 8.06 | 2.57 | 2.38 | 52.99 | 11.99 | 31.07 | 3.78 | 34.83 | 4.80 | 30.08 | 17.51 | 7.46 | 5.86 | 5.48 | 2.70 | - | - | - | - |
| | OccFormer | C | 40.27 | 13.81 | 22.58 | 0.66 | 0.26 | 9.89 | 3.82 | 2.77 | 54.3 | 13.44 | 31.53 | 3.55 | 36.42 | 4.80 | 31.00 | 19.51 | 7.77 | 8.51 | 6.95 | 4.60 | - | - | - | - |
| SSCBench-nuScenes | LMSCNet | L | 21.09 | 8.36 | 14.74 | 0 | 0 | 5.93 | 8.52 | 3.41 | 24.14 | - | 7.55 | 8.40 | 18.56 | - | 9.02 | - | - | - | - | 0 | - | - | - |
| | SSCNet | L | 27.64 | 11.84 | 18.06 | 0 | 0.36 | 13.42 | 10.35 | 7.59 | 28.74 | - | 12.65 | 12.65 | 21.05 | - | 16.33 | - | - | - | - | 0.01 | - | - | - |
| | MonoScene | C | 29.63 | 9.60 | 10.17 | 1.7 | 3.80 | 8.35 | 8.74 | 3.72 | 38.77 | - | 14.74 | 12.58 | 7.23 | - | 5.50 | - | - | - | - | 0.03 | - | - | - |
| | Voxformer | C | 25.16 | 4.96 | 4.95 | 0.29 | 1.21 | 2.73 | 2.45 | 1.12 | 23.94 | - | 10.14 | 4.06 | 3.97 | - | 4.58 | - | - | - | - | 0.06 | - | - | - |
| | OccFormer | C | 28.23 | 7.55 | 14.61 | 2.25 | 7.97 | 11.88 | 9.80 | 5.87 | 37.62 | - | 18.63 | 19.76 | 9.05 | - | 5.92 | - | - | - | - | 0 | - | - | - |
| SSCBench-Waymo | LMSCNet | L | 89.04 | 38.58 | 75.21 | 0 | 0 | - | - | 58.22 | 76.29 | - | 51.07 | 51.30 | 71.63 | - | 73.30 | - | 41.58 | 27.24 | - | 8.83 | 0.19 | 9.12 | 34.69 |
| | SSCNet | L | 85.43 | 41.63 | 69.97 | 13.23 | 0 | - | - | 58.32 | 75.40 | - | 51.63 | 52.12 | 70.15 | - | 70.93 | - | 41.88 | 33.07 | - | 23.28 | 15.37 | 13.15 | 35.89 |
| | MonoScene | C | 36.81 | 10.54 | 14.54 | 1.36 | 0.03 | - | - | 6.59 | 42.51 | - | 19.82 | 18.96 | 12.55 | - | 11.48 | - | 3.48 | 3.59 | - | 2.94 | 3.01 | 14.43 | 2.88 |
| | Voxformer | C | 36.36 | 9.46 | 14.58 | 0.36 | 0.02 | - | - | 5.20 | 40.79 | - | 17.26 | 15.68 | 10.33 | - | 11.34 | - | 3.10 | 2.71 | - | 2.38 | 1.83 | 13.86 | 2.49 |
| | TPVFormer | C | 36.78 | 10.91 | 15.23 | 1.54 | 0.23 | - | - | 4.88 | 43.02 | - | 22.31 | 19.78 | 13.53 | - | 12.75 | - | 2.96 | 2.53 | - | 4.65 | 2.90 | 14.78 | 2.53 |

# 5 SEPARATE BENCHMARKING RESULTS

## 5.1 QUANTITATIVE COMPARISONS

**Camera-based Methods.** On SSCBench-KITTI-360, TPVFormer and OccFormer demonstrate superior geometry completion performance (IoU) compared to VoxFormer and MonoScene, as illustrated in Tab. 2. Improved geometry completion contributes to enhancing semantic segmentation (mIoU). On SSCBench-Waymo, MonoScene marginally outperforms VoxFormer across the majority of evaluation metrics. Due to the absence of stereo data in SSCBench-Waymo, the utilization of off-the-shelf self-supervised depth estimation modules (Bhat et al., 2021; Shamsafar et al., 2022), primarily trained on KITTI, results in suboptimal depth knowledge, leading to less competitive performance by VoxFormer in SSC. This trend becomes more pronounced in SSCBench-nuScenes, where the IoU and mIoU metrics for MonoScene significantly surpass those of VoxFormer (IoU, $29.63 \rightarrow 25.16$ and mIoU, $9.60 \rightarrow 4.96$). It is evident that accurate depth estimation plays a crucial role in scene geometry estimation within camera-based methods.

**LiDAR-based Methods.** SSCNet consistently outperforms LMSCNet across all three subsets, mainly due to its larger number of parameters (1.03M compared to 0.35M). Additionally, it is worth noting that SSCNet exhibits superior recognition capabilities for small objects, such as motorcycles (■, $2.24 \leftrightarrow 0.00$ in SSCBench-nuScenes) and bicycles (■, $13.23 \leftrightarrow 0.00$ in SSCBench-Waymo). This demonstrates SSCNet's advantage in handling sparse LiDAR data compared to LMSCNet. When comparing results between SSCBench-KITTI-360 and SSCBench-nuScenes to SSCBench-Waymo, SSCNet and LMSCNet consistently deliver significantly better performance on the SSCBench-Waymo dataset, with IoU values approaching 90%. This improvement can be attributed to the denser LiDAR input available in Waymo data. However, it is crucial to emphasize that while dense LiDAR input leads to satisfactory performance, implementing this 5-LiDAR setup remains costly for most common autonomous driving solutions.

**Camera *vs.* LiDAR.** As demonstrated in Tab. 2, on SSCBench-KITTI-360, LiDAR-based methods outperform camera-based approaches in terms of geometry metrics and most semantic metrics. This outcome is expected since camera-based methods must infer 3D scene geometry from 2D images, while LiDAR-based methods directly extract scene geometry from LiDAR input. However, the scenario changes in SSCBench-nuScenes, where camera-based methods generally surpass LiDAR-based methods in terms of IoU. This difference can be attributed to the use of a sparse LiDAR sensor (Velodyne HDL32E) in the nuScenes dataset. These results indicate that LiDAR-based methods are sensitive to the sparsity of input. Specifically, while dense input has the potential for significant performance improvement, sparse input can lead to significant degradation. This observation is further confirmed in SSCBench-Waymo, where the Waymo dataset contains point cloud data collected from

Table 3: **Comparison between monocular and trinocular setup on SSCBench-Waymo**. The monocular setup utilizes the front camera only. The trinocular setup utilizes images from all 3 front-facing cameras, which collectively provide a 180° view.

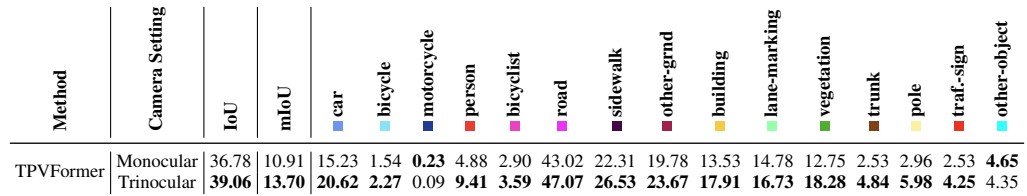

| Method | Camera Setting | IoU | mIoU | ■ car | ■ bicycle | ■ motorcycle | ■ person | ■ bicyclist | ■ road | ■ sidewalk | ■ other-grnd | ■ building | ■ lane-marking | ■ vegetation | ■ trunk | ■ pole | ■ traf.-sign | ■ other-object |
|---|---|---|---|---|---|---|---|---|---|---|---|---|---|---|---|---|---|---|
| TPVFormer | Monocular | 36.78 | 10.91 | 15.23 | 1.54 | **0.23** | 4.88 | 2.90 | 43.02 | 22.31 | 19.78 | 13.53 | 14.78 | 12.75 | 2.53 | 2.96 | 2.53 | **4.65** |
| | Trinocular | **39.06** | **13.70** | **20.62** | **2.27** | 0.09 | **9.41** | **3.59** | **47.07** | **26.53** | **23.67** | **17.91** | **16.73** | **18.28** | **4.84** | **5.98** | **4.25** | 4.35 |

one mid-range and four short-range LiDARs. As seen in Tab. 2, the two LiDAR-based methods outperform camera-based methods by a significant margin on all metrics in SSCBench-Waymo.

However, camera-based methods outperform LiDAR-based ones for smaller objects that comprise a minuscule fraction of samples ($< 0.5\%$). For classes such as bicycles (■, $0.00 \rightarrow 1.16$), persons (■, $0.26 \rightarrow 4.54$), poles (■, $1.09 \rightarrow 12.93$), and traffic signs (■, $0.90 \rightarrow 14.25$) in SSCBench-KITTI-360, as well as motorcycles (■, $0.00 \rightarrow 3.80$) and bicycles (■, $0.00 \rightarrow 1.70$) in SSCBench-nuScenes, camera-based methods significantly outperform LiDAR-based methods. Despite the low frequency of small objects, their identification is vitally important for collision avoidance and traffic understanding.

## 5.2 DISCUSSIONS AND ANALYSES

**Impact of Point Cloud Density.** Our experiments illuminate the impact of LiDAR input density on model performance. In the SSCBench-nuScenes dataset, which features relatively sparse LiDAR input (32 channels), camera-based methods outperform LiDAR-based methods on geometric metrics. However, in the SSCBench-Waymo dataset, which benefits from dense LiDAR input (64 channels, 5 LiDARs), LiDAR-based methods vastly outperform camera-based methods. The sensitivity of LiDAR-based methods to input becomes evident, with advantages observed in dense input and notable performance degradation in sparse input. This highlights the need for future research in developing robust LiDAR-based methods that can mitigate degradation while capitalizing on the benefits.

**Monocular vs. Trinocular.** Table 3 displays the performance of TPVFormer with monocular and trinocular input. While a trinocular setup offers a broader field of view that can help enhance overall performance in terms of both IoU ($36.78 \rightarrow 39.06$) and mIoU ($10.91 \rightarrow 13.70$), achieving excellent results using only a single camera remains a compelling academic challenge. There is still significant research value in developing monocular methods that can match the performance of models with panoramic views, as they are memory-efficient, computation-efficient, and easy to deploy.

**Comparison with SemanticKITTI.** We observe significant discrepancies when comparing our experimental results on SSCBench to those from SemanticKITTI (Behley et al., 2019) (for more details, we refer readers to VoxFormer (Li et al., 2023)). While VoxFormer performs admirably well on SemanticKITTI in metrics such as IoU and mIoU, it faces challenges with the diversity of our SSCBench dataset. This challenge primarily arises from its depth estimation module's inability to generalize beyond SemanticKITTI. Furthermore, LMSCNet, which typically exhibits superior geometric performance compared to SSCNet on SemanticKITTI, demonstrates the opposite trend on SSCBench. These discrepancies underscore two essential points. First, they highlight the significance of SSCBench, which provides diverse and demanding real-world scenarios for comprehensive evaluations. Second, they emphasize the necessity for robust methods capable of maintaining high performance across various environments.

## 6 UNIFIED BENCHMARKING RESULTS

To assess the domain gap and compare the cross-domain generalizability of state-of-the-art algorithms, we established a unified benchmark for cross-validation on SSCBench. Specifically, we employed two LiDAR-based methods, LMSCNet (Roldao et al., 2020) and SSCNet (Song et al., 2017), and one camera-based method, MonoScene (Cao & de Charette, 2022), for experiments on SSCBench-KITTI-360 and SSCBench-Waymo. To ensure consistent evaluation metrics, we standardized the labels of

Table 4: **Cross-domain evaluation**. We report the experiment results of training and testing on different datasets using unified labels.

| Training-dataset | Testing-dataset | Method | IoU | mIoU | vehicle | bicycle | motorcycle | person | road | sidewalk | other-grnd | building | vegetation | other-object |
|---|---|---|---|---|---|---|---|---|---|---|---|---|---|---|
| SSCBench-KITTI-360 | SSCBench-KITTI-360 | LMSCNet | 48.49 | 22.48 | 23.87 | 0.00 | 0.00 | 0.00 | 64.04 | 35.13 | 17.46 | 44.47 | 39.755 | 0.04 |
| | | SSCNet | **54.50** | **25.63** | **34.14** | 0.00 | 0.00 | 0.11 | 67.54 | **41.84** | **19.86** | **46.89** | **44.92** | 1.01 |
| | | MonoScene | 39.11 | 19.53 | 21.29 | **2.60** | **1.60** | **3.27** | 51.62 | 30.42 | 14.25 | 34.83 | 29.16 | **6.26** |
| | SSCBench-Waymo | LMSCNet | **17.45** | 1.83 | 0.36 | 0.00 | 0.00 | 0.00 | **4.02** | 0.82 | **2.77** | 2.74 | 9.99 | 0.00 |
| | | SSCNet | 17.82 | **4.18** | 3.11 | 0.00 | 0.00 | 0.00 | 3.47 | **1.47** | 2.31 | **18.20** | **12.77** | **0.44** |
| | | MonoScene | 7.07 | 2.16 | **9.87** | **0.01** | 0.00 | **2.12** | 2.99 | 0.55 | 0.24 | 0.45 | 5.00 | 0.38 |
| SSCBench-Waymo | SSCBench-KITTI-360 | LMSCNet | 4.12 | 1.04 | 1.34 | 0.00 | 0.00 | 0.03 | 1.65 | 0.19 | 0.90 | 4.03 | 2.19 | 0.06 |
| | | SSCNet | 8.99 | 1.33 | **2.73** | 0.00 | 0.00 | **0.04** | 1.73 | **0.27** | 0.50 | **5.53** | 2.46 | **0.07** |
| | | MonoScene | **14.72** | **2.28** | 0.98 | 0.00 | 0.00 | 0.00 | **6.58** | 0.19 | **2.37** | 5.11 | **5.26** | 0.02 |
| | SSCBench-Waymo | LMSCNet | **89.59** | **51.98** | **76.56** | 0.00 | 0.00 | **62.69** | **79.96** | 52.37 | 52.27 | **72.45** | **74.94** | 48.54 |
| | | SSCNet | 85.89 | 51.31 | 71.55 | **5.65** | 0.00 | 60.26 | 78.65 | 53.11 | **53.02** | 70.17 | 71.99 | **48.73** |
| | | MonoScene | 54.61 | 14.67 | 16.09 | 2.03 | **0.09** | 6.82 | 46.31 | 23.14 | 20.92 | 13.70 | 13.90 | 3.65 |

SSCBench-KITTI-360 and SSCBench-Waymo to a unified set comprising 10 common objects. All other experimental settings and evaluation metrics adhere to the guidelines outlined in Sec. 4.

**Overall Performance.** As shown in Tab. 4, all three methods exhibit a notable decline in performance when cross-validated on another dataset across both the geometric metric (IoU) as well as the semantic metric (mIoU), regardless of the training dataset. Specifically, the model trained on SSCBench-Waymo and tested on SSCBench-KITTI-360 suffers a more severe decline for LiDAR-based methods than the other way around. This is because SSCBench-Waymo has a very dense point cloud input from five LiDARs, which effectively reduces the performance degradation caused by domain differences. Interestingly, the deterioration trend in terms of mIoU for MonoScene is more severe when transferring from SSCBench-KITTI-360 to SSCBench-Waymo than the other way around. This can be partially explained by the higher in-domain mIoU on SSCBench-KITTI-360 than that on SSCBench-Waymo and the difference in input resolutions ($1408 \times 376 \leftrightarrow 960 \times 640$), which is magnified by the fixed model parameters, and thereby affects feature representation.

**Class-Specific Performance.** We observe the most significant performance drop in the "road" class (■) for both transfer directions in all three methods. This suggests that ground types may be represented differently across datasets, causing challenges for both camera-based and LiDAR-based methods to adapt to these variations. It is worth noting that after unifying the labels in SSCBench-KITTI-360, the score for "motorcycle" (■) drops to 0 (from 1.02). This demonstrates that a relatively smaller number of classes can impact feature extraction and classification, particularly for uncommon small objects. Throughout cross-domain evaluations, a recurring phenomenon of deterioration can be observed. This phenomenon underscores the value and necessity of our proposed SSCBench dataset. It is envisioned that models trained on this dataset would be better equipped to handle the variations and complexities encountered in cross-domain scenarios. Moreover, it also serves as motivation for the development of models that are more robust and capable of generalizing across different domains.

# 7 CONCLUSION

**Limitations and Future Work.** SSCBench only encompasses 3D data following the convention of the SSC problem. This limits evaluations of 4D methods with temporal dimensions. Future work will aim to expand SSCBench to include temporal information.

**Summary.** In this paper, we introduce SSCBench, a large-scale benchmark composed of diverse street views, aimed at facilitating the development of robust and generalizable semantic scene completion models. Through meticulous curation and comprehensive benchmarking, we identify the bottlenecks of existing methods and offer valuable insights into future research directions. Our ambition is for SSCBench to stimulate advancements in 3D semantic scene completion, ultimately enhancing perception capabilities for the next-generation autonomous systems.

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

APPENDIX

## A  OVERVIEW

We present the necessary licenses in Appendix B, because our benchmark is constructed based on the previous works (Liao et al., 2022; Caesar et al., 2020; Sun et al., 2020). We also exhaustively analyze the impact of perception distance on the evaluation performance in Appendix C, where we present detailed results comparison from short-range to long-range under different settings. In Appendix D, we display representative qualitative visualization results from our SSCBench, along with the input image and LiDAR. Finally, the visualization comparison between the monocular method and trinocular method has also been presented in Appendix D for better qualitative evaluation.

## B  DATASET DETAILS

Complying with the original dataset we used, different parts of the SSCBench dataset are released under different licenses. In particular:

- SSCBench-KITTI-360: CC BY-NC-SA 3.0
- SSCBench-nuScenes: CC BY-NC-SA 4.0
- SSCBench-Waymo: Waymo Dataset License Agreement for Non-Commercial Use

The authors bear all responsibilities for the licensing, distribution, and maintenance of our datasets.

To improve the usability and readability of our SSCBench, we will release the complete dataset and detailed configurations upon acceptance, along with the codes and corresponding models for the evaluation methods on our SSCBench, based on their official codebases.

## C  ANALYSIS OF RANGE

We report the performances across three distinct scales of volume, namely $12.8 \times 12.8 \times 6.4\text{m}^3$, $25.6 \times 25.6 \times 6.4\text{m}^3$, and $51.2 \times 51.2 \times 6.4\text{m}^3$. These scales have been deliberately selected and provide an in-depth analysis of performance across varying perception ranges. Such an understanding is paramount for a variety of subsequent tasks, especially those associated with vehicle planning and navigation (Hu et al., 2023). Notably, all the aforementioned scales lie within the observable range of onboard sensors, such as cameras and LiDAR. The design choice of our evaluation from short-range to long-range underscores its significance in assessing the vehicle's forward anticipation and the associated implications on performance in relation to the perception range.

### C.1  PER SUBSET

In Tab. I, II, and III, comprehensive performances of the proposed three subsets SSCBench-KITTI-360, SSCBench-nuScenes, and SSCBench-Waymo have been demonstrated, especially for the three distinct evaluation ranges. As the perception range expands, all LiDAR-based and camera-based methods generally experience a decrease in performance due to increased complexity and uncertainty in larger volumes. However, LiDAR-based methods, specifically in SSCBench-Waymo, demonstrate minimal performance degradation. This trend, likely attributed to the dense LiDAR input, highlights their ability to maintain stable performance across varying perception ranges. It underscores the potential benefits of dense LiDAR data collection, albeit with increased cost and computational demand. Additionally, this trend emphasizes the need for further research to enhance LiDAR data density and advance depth estimation for camera-based methods, improving their scalability to larger perception ranges.

### C.2  TRINOCULAR *vs.* MONOCULAR

Mirroring the degradation with SSCBench-Waymo, trinocular setting can also confront a decrease in performance along with the increase of evaluation range, shown in Tab. IV. However, it is worth noting that while trinocular method exhibits a comparable decline in geometric metrics (IoU) to that of monocular method, it experiences a more pronounced deterioration in semantic metrics (mIoU). One possible explanation might be that trinocular method becomes more reliant on semantic information due to the influx of additional visual information from triple inputs. As the evaluation range expands,

Table I: **Quantitative comparison on SSCBench-KITTI-360**. We report the performances with respect to different ranges (12.8m, 25.6m, and 51.2m). We provide both geometric (IoU) and semantic (mIoU) metrics. The top three performances are marked by **red**, **green**, and **blue** respectively.

| Methods | TPVFormer | | | OccFormer | | | VoxFormer-S | | | MonoScene | | | LMSCNet | | | SSCNet | | |
|---|---|---|---|---|---|---|---|---|---|---|---|---|---|---|---|---|---|---|
| Range (m) | 12.8 | 25.6 | 51.2 | 12.8 | 25.6 | 51.2 | 12.8 | 25.6 | 51.2 | 12.8 | 25.6 | 51.2 | 12.8 | 25.6 | 51.2 | 12.8 | 25.6 | 51.2 |
| IoU (%) | 56.56 | 46.83 | 40.22 | 58.71 | 47.96 | 40.27 | 55.45 | 46.36 | 38.76 | 54.65 | 44.70 | 37.87 | 65.34 | 57.43 | 47.35 | 70.97 | 62.96 | 53.58 |
| Precision (%) | 68.14 | 62.41 | 59.32 | 69.47 | 62.68 | 59.7 | 66.10 | 61.34 | 58.52 | 65.88 | 59.96 | 56.73 | 77.14 | 73.07 | 72.77 | 78.17 | 72.25 | 69.63 |
| Recall (%) | 76.90 | 65.21 | 55.54 | 79.13 | 67.12 | 55.31 | 77.48 | 65.48 | 53.44 | 76.24 | 63.72 | 53.26 | 81.03 | 72.85 | 57.55 | 88.52 | 83.05 | 69.92 |
| mIoU | 22.50 | 18.03 | 13.64 | 23.04 | 18.38 | 13.81 | 18.17 | 15.4 | 11.91 | 20.29 | 16.18 | 12.31 | 20.04 | 17.57 | 13.65 | 24.01 | 21.40 | 16.95 |
| ■ car (2.85%) | 36.70 | 30.60 | 21.56 | 40.87 | 33.1 | 22.58 | 29.41 | 25.08 | 17.84 | 30.83 | 26.35 | 19.34 | 37.49 | 31.71 | 20.91 | 50.02 | 44.47 | 31.95 |
| ■ bicycle (0.01%) | 3.40 | 2.07 | 1.09 | 1.94 | 1.04 | 0.66 | 2.73 | 1.73 | 1.16 | 1.94 | 0.83 | 0.43 | 0.00 | 0.00 | 0.00 | 0.00 | 0.00 | 0.00 |
| ■ motorcycle (0.01%) | 5.17 | 2.51 | 1.37 | 1.03 | 0.43 | 0.26 | 1.97 | 1.47 | 0.89 | 3.25 | 1.30 | 0.58 | 0.00 | 0.00 | 0.00 | 1.02 | 0.36 | 0.17 |
| ■ truck (0.16%) | 17.72 | 12.94 | 8.06 | 22.4 | 15.21 | 9.89 | 6.08 | 6.63 | 4.56 | 14.83 | 12.18 | 8.02 | 0.65 | 0.46 | 0.26 | 14.67 | 13.69 | 10.29 |
| ■ other-veh. (5.75%) | 5.78 | 4.74 | 2.57 | 8.48 | 6.12 | 3.82 | 3.71 | 3.56 | 2.06 | 6.08 | 4.30 | 2.03 | 1.49 | 1.01 | 0.58 | 0.00 | 0.00 | 0.00 |
| ■ person (0.02%) | 3.60 | 3.19 | 2.38 | 4.54 | 3.79 | 2.77 | 2.86 | 2.20 | 1.63 | 2.06 | 1.26 | 0.86 | 0.00 | 0.00 | 0.00 | 0.26 | 0.13 | 0.07 |
| ■ road (14.98%) | 73.31 | 65.73 | 52.99 | 73.34 | 66.53 | 54.3 | 66.10 | 58.58 | 47.01 | 68.60 | 59.93 | 48.35 | 77.35 | 74.32 | 62.95 | 78.03 | 75.96 | 65.70 |
| ■ parking (2.31%) | 28.38 | 18.35 | 11.99 | 30.91 | 20.25 | 13.44 | 18.44 | 13.52 | 9.67 | 24.32 | 16.4 | 11.38 | 25.56 | 20.43 | 13.51 | 29.60 | 24.51 | 17.33 |
| ■ sidewalk (6.43%) | 48.06 | 39.79 | 31.07 | 49.76 | 41.3 | 31.53 | 38.00 | 33.63 | 27.21 | 44.43 | 36.05 | 28.13 | 51.20 | 44.42 | 33.51 | 56.60 | 51.40 | 41.24 |
| ■ other-grnd (2.05%) | 6.07 | 5.28 | 3.78 | 5.62 | 5.45 | 3.55 | 4.49 | 4.04 | 2.89 | 5.76 | 4.82 | 3.32 | 0.41 | 0.32 | 0.20 | 6.88 | 5.36 | 3.22 |
| ■ building (15.67%) | 50.45 | 43.11 | 34.83 | 53.65 | 44.86 | 36.42 | 41.12 | 38.24 | 31.18 | 45.40 | 40.60 | 32.89 | 58.66 | 54.08 | 43.67 | 62.14 | 54.04 | 44.41 |
| ■ fence (0.96%) | 9.40 | 7.68 | 4.8 | 10.64 | 7.85 | 4.8 | 8.99 | 7.43 | 4.97 | 9.79 | 5.91 | 3.53 | 1.66 | 0.71 | 0.33 | 13.53 | 11.06 | 6.77 |
| ■ vegetation (41.99%) | 45.49 | 35.73 | 30.08 | 49.91 | 37.96 | 31 | 45.68 | 35.16 | 28.99 | 42.98 | 32.75 | 26.15 | 55.87 | 48.52 | 40.01 | 59.28 | 51.96 | 43.72 |
| ■ terrain (7.10%) | 31.25 | 22.02 | 17.52 | 34.63 | 24.99 | 19.51 | 24.70 | 18.53 | 14.69 | 31.96 | 21.63 | 16.75 | 41.96 | 34.15 | 26.80 | 40.80 | 34.84 | 28.87 |
| ■ pole (0.22%) | 11.82 | 9.74 | 7.46 | 12.93 | 10.25 | 7.77 | 8.84 | 8.16 | 6.51 | 9.28 | 8.45 | 6.51 | 0.00 | 0.00 | 0.00 | 1.09 | 1.19 | 0.78 |
| ■ traf.-sign (0.06%) | 11.17 | 8.49 | 5.86 | 14.25 | 12.37 | 8.51 | 9.15 | 9.02 | 6.92 | 8.58 | 7.67 | 5.67 | 0.00 | 0.00 | 0.00 | 0.90 | 1.16 | 0.75 |
| ■ other-struct. (4.33%) | 11.72 | 8.98 | 5.48 | 13.81 | 11.04 | 6.95 | 10.31 | 7.02 | 3.79 | 9.18 | 6.76 | 4.20 | 9.87 | 7.09 | 3.63 | 14.80 | 12.98 | 8.60 |
| ■ other-obejct (0.28%) | 5.48 | 3.59 | 2.70 | 8.96 | 6.71 | 4.6 | 4.40 | 3.27 | 2.43 | 5.86 | 4.49 | 3.09 | 0.0028 | 0.0007 | 0.0003 | 1.13 | 1.07 | 0.67 |

Table II: **Quantitative comparison on SSCBench-nuScenes**. All metrics and legends follow those in Tab. I.

| Methods | OccFormer | | | VoxFormer-S | | | MonoScene | | | LMSCNet | | | SSCNet | | |
|---|---|---|---|---|---|---|---|---|---|---|---|---|---|---|---|
| Range | 12.8m | 25.6m | 51.2m | 12.8m | 25.6m | 51.2m | 12.8m | 25.6m | 51.2m | 12.8m | 25.6m | 51.2m | 12.8m | 25.6m | 51.2m |
| IoU (%) | 62.48 | 42.94 | 28.23 | 59.75 | 39.25 | 25.16 | 65.06 | 44.79 | 29.63 | 61.40 | 36.98 | 21.09 | 63.44 | 43.06 | 27.64 |
| Precision (%) | 86.77 | 73.99 | 60.08 | 85.48 | 72.33 | 53.11 | 88.83 | 75.95 | 60.64 | 92.89 | 86.48 | 83.12 | 84.22 | 68.89 | 59.21 |
| Recall (%) | 69.05 | 50.57 | 34.75 | 66.5 | 46.18 | 32.34 | 70.84 | 52.09 | 36.69 | 64.03 | 39.25 | 22.03 | 72.00 | 53.45 | 34.14 |
| mIoU | 14.24 | 10.75 | 7.55 | 8.89 | 7.05 | 4.96 | 18.89 | 13.59 | 9.60 | 22.4 | 14.19 | 8.36 | 25.7 | 18.24 | 11.84 |
| ■ car (2.47%) | 28.20 | 21.29 | 14.61 | 8.23 | 7.65 | 4.95 | 21.95 | 14.86 | 10.17 | 38.62 | 24.29 | 14.74 | 38.36 | 27.04 | 18.06 |
| ■ bicycle (0.16%) | 6.97 | 3.57 | 2.25 | 0.26 | 0.38 | 0.29 | 4.13 | 2.39 | 1.70 | 0.00 | 0.00 | 0.00 | 0.00 | 0.00 | 0.00 |
| ■ motorcycle (0.02%) | 12.22 | 10.29 | 7.97 | 1.56 | 1.99 | 1.21 | 5.86 | 5.72 | 3.80 | 0.00 | 0.00 | 0.00 | 2.24 | 0.77 | 0.36 |
| ■ truck (0.96%) | 22.26 | 17.05 | 11.88 | 4.06 | 4.05 | 2.73 | 17.18 | 11.55 | 8.35 | 18.40 | 10.32 | 5.93 | 26.67 | 18.89 | 13.42 |
| ■ other-veh. (7.87%) | 18.84 | 15.31 | 9.80 | 5.17 | 4.13 | 2.45 | 18.51 | 11.87 | 8.74 | 17.58 | 13.28 | 8.52 | 20.01 | 14.85 | 10.35 |
| ■ person (0.24%) | 16.08 | 8.74 | 5.87 | 2.53 | 1.71 | 1.12 | 10.65 | 5.66 | 3.72 | 13.22 | 6.37 | 3.41 | 18.15 | 12.25 | 7.59 |
| ■ road (36.97%) | 70.52 | 54.74 | 37.62 | 48.79 | 36.19 | 23.94 | 70.60 | 54.59 | 38.77 | 63.39 | 40.82 | 24.14 | 65.28 | 45.49 | 28.74 |
| ■ sidewalk (8.30%) | 37.57 | 28.43 | 18.63 | 17.22 | 14.45 | 10.14 | 32.75 | 23.03 | 14.74 | 26.10 | 14.29 | 7.55 | 35.46 | 23.23 | 12.65 |
| ■ other-grnd (0.75%) | 35.58 | 27.39 | 19.76 | 8.56 | 6.52 | 4.06 | 28.86 | 19.94 | 12.58 | 26.90 | 15.29 | 8.40 | 34.63 | 22.00 | 12.65 |
| ■ building (21.17%) | 10.30 | 9.77 | 9.05 | 2.62 | 2.30 | 3.97 | 5.28 | 6.51 | 7.23 | 36.77 | 28.94 | 18.56 | 35.83 | 30.44 | 21.05 |
| ■ vegetation (20.97%) | 12.09 | 7.63 | 5.92 | 7.62 | 5.18 | 4.58 | 10.79 | 6.90 | 5.50 | 27.87 | 16.74 | 9.02 | 31.81 | 23.91 | 16.33 |
| ■ other-obejct (0.11%) | 0.00 | 0.00 | 0.00 | 0.02 | 0.05 | 0.06 | 0.29 | 0.06 | 0.03 | 0.00 | 0.00 | 0.00 | 0.00 | 0.02 | 0.01 |

there is a potential misalignment in the features from the visual data captured by the cameras heading to different view angles, which can have a more adverse effect on the recognition of distant objects.

## C.3 UNIFIED BENCHMARK RESULTS

Table V, VI, and VII demonstrate the cross-domain evaluation across three different evaluation ranges, following the unified setting mentioned above. It is noteworthy that for LMSCNet, after training on SSCBench-KITTI-360 and testing on SSCBench-Waymo, there is a subtle improvement in performance as the evaluation distance increases. This is likely attributed to the denser LiDAR data in the SSCBench-Waymo. While a similar trend can be observed for SSCNet, a distinction emerges when trained on SSCBench-Waymo and tested on SSCBench-360: SSCNet demonstrates greater robustness to the change of distance compared to LMSCNet, which is likely due to the aforementioned number of parameters. Interestingly, for MonoScenes, the model showcases an insensitivity to the evaluation range across different datasets. Specifically, the semantic metric (mIoU) for MonoScenes experiences a decline of less than 1%.

## D QUALITATIVE RESULTS

In Fig. I, we show the completion results of different methods on the SSCBench dataset. We can observe comparable results between LMSCNet (Roldao et al., 2020) and SSCNet (Song et al., 2017), while MonoScene (Cao & de Charette, 2022) has a varying performance depending on the input. On SSCBench-KITTI-360, MonoScene has comparable results to the two LiDAR-based methods due to the large FOV. It also shows a superior completion ability when there is occlusion near the ego-vehicle

Table III: **Quantitative comparison on SSCBench-Waymo**. All metrics and legends follow Tab. I.

| Methods | TPVFormer | | | VoxFormer-S | | | MonoScene | | | LMSCNet | | | SSCNet | | |
|---|---|---|---|---|---|---|---|---|---|---|---|---|---|---|---|
| Range | 12.8m | 25.6m | 51.2m | 12.8m | 25.6m | 51.2m | 12.8m | 25.6m | 51.2m | 12.8m | 25.6m | 51.2m | 12.8m | 25.6m | 51.2m |
| IoU (%) | 61.59 | 45.07 | 36.78 | 60.36 | 44.09 | 36.36 | 62.44 | 45.36 | 36.81 | 89.19 | 89.19 | 89.04 | 85.26 | 85.34 | 85.43 |
| Precision (%) | 75.65 | 60.57 | 51.53 | 75.95 | 62.90 | 52.00 | 79.03 | 63.16 | 52.11 | 94.83 | 94.84 | 94.74 | 88.62 | 88.71 | 88.83 |
| Recall (%) | 76.81 | 63.78 | 56.24 | 75.03 | 59.58 | 54.72 | 74.84 | 61.67 | 55.62 | 93.75 | 93.73 | 93.67 | 95.74 | 95.74 | 95.7 |
| mIoU | 13.85 | 13.16 | 10.91 | 9.65 | 10.11 | 9.46 | 12.39 | 12.26 | 10.54 | 39.10 | 39.05 | 38.58 | 42.37 | 42.17 | 41.63 |
| ■ car (7.65%) | 20.57 | 19.01 | 15.24 | 14.60 | 17.13 | 14.58 | 18.18 | 17.82 | 14.54 | 74.13 | 74.99 | 75.21 | 68.05 | 69.36 | 69.97 |
| ■ bicycle (0.02%) | 0.89 | 1.52 | 1.54 | 0.13 | 0.38 | 0.36 | 0.91 | 0.72 | 1.36 | 0.00 | 0.00 | 0.00 | 17.04 | 14.70 | 13.23 |
| ■ motorcycle (0.001%) | 0.03 | 0.10 | 0.23 | 0.00 | 0.05 | 0.02 | 0.04 | 0.14 | 0.03 | 0.00 | 0.00 | 0.00 | 0.00 | 0.00 | 0.00 |
| ■ person (0.84%) | 9.14 | 6.34 | 4.88 | 8.59 | 6.61 | 5.20 | 12.76 | 9.23 | 6.59 | 61.47 | 59.75 | 58.22 | 61.13 | 60.00 | 58.32 |
| ■ bycyclist (0.03%) | 2.08 | 4.91 | 2.90 | 0.95 | 2.73 | 1.83 | 0.61 | 3.23 | 3.01 | 0.16 | 0.26 | 0.19 | 15.13 | 15.81 | 15.37 |
| ■ road (29.62%) | 64.75 | 53.21 | 43.02 | 63.07 | 49.46 | 40.79 | 66.34 | 53.71 | 42.51 | 76.19 | 76.77 | 76.29 | 75.07 | 75.65 | 75.40 |
| ■ sidewalk (6.65%) | 31.96 | 28.02 | 22.31 | 18.77 | 18.01 | 17.26 | 26.57 | 24.14 | 19.82 | 51.53 | 50.96 | 51.07 | 52.07 | 51.75 | 51.63 |
| ■ other-grnd(11.40%) | 22.16 | 21.65 | 19.78 | 8.05 | 13.17 | 15.68 | 17.45 | 20.42 | 18.96 | 51.29 | 51.49 | 51.30 | 51.90 | 52.15 | 52.12 |
| ■ building (18.38%) | 10.30 | 14.75 | 13.53 | 1.71 | 5.59 | 10.33 | 7.78 | 13.48 | 12.55 | 72.08 | 72.01 | 71.63 | 70.61 | 70.51 | 70.15 |
| ■ lane-marking (1.15%) | 20.47 | 18.10 | 14.78 | 11.93 | 15.42 | 13.86 | 12.19 | 15.51 | 14.43 | 10.63 | 10.19 | 9.12 | 14.67 | 14.33 | 13.15 |
| ■ vegetation (22.41%) | 14.28 | 13.24 | 12.75 | 9.72 | 9.72 | 11.34 | 10.03 | 10.18 | 11.48 | 73.67 | 73.55 | 73.30 | 71.06 | 71.03 | 70.93 |
| ■ trunk (0.90%) | 2.69 | 3.31 | 2.53 | 2.14 | 2.95 | 2.49 | 3.35 | 3.52 | 2.88 | 35.01 | 35.06 | 34.69 | 35.80 | 35.83 | 35.89 |
| ■ pole (0.72%) | 1.77 | 3.18 | 2.96 | 1.52 | 3.19 | 3.10 | 2.88 | 3.79 | 3.48 | 42.23 | 42.47 | 41.58 | 42.34 | 42.21 | 41.88 |
| ■ traf.-sign (0.19%) | 1.71 | 2.32 | 2.53 | 2.06 | 3.08 | 2.71 | 4.13 | 3.74 | 3.59 | 28.63 | 28.78 | 27.24 | 35.21 | 34.51 | 33.07 |
| ■ other-obejct (0.03%) | 4.99 | 7.70 | 4.65 | 1.50 | 4.18 | 2.38 | 2.7 | 4.21 | 2.94 | 9.54 | 9.46 | 8.83 | 25.46 | 24.65 | 23.28 |

Table IV: **Comparison between monocular and trinocular setup on SSCBench-Waymo**. In the monocular setup, we only use front camera images. However, in the trinocular setup, we utilize 3 camera images, which collectively provide a 180° view.

| Methods | TPVFormer-Mono | | | TPVFormer-Tri | | |
|---|---|---|---|---|---|---|
| Range | 12.8m | 25.6m | 51.2m | 12.8m | 25.6m | 51.2m |
| IoU (%) | 61.59 | 45.07 | 36.78 | 65.03 | 49.38 | 39.06 |
| Precision (%) | 75.65 | 60.57 | 51.53 | 76.76 | 64.40 | 54.65 |
| Recall (%) | 76.81 | 63.78 | 56.24 | 80.98 | 67.92 | 57.79 |
| mIoU | 13.85 | 13.16 | 10.91 | 20.62 | 18.07 | 13.71 |
| ■ car (7.65%) | 20.57 | 19.01 | 15.24 | 35.56 | 28.57 | 20.62 |
| ■ bicycle (0.02%) | 0.89 | 1.52 | 1.54 | 6.58 | 3.21 | 2.27 |
| ■ motorcycle (0.001%) | 0.03 | 0.10 | 0.23 | 0.29 | 0.18 | 0.09 |
| ■ person (0.84%) | 9.14 | 6.34 | 4.88 | 21.74 | 14.89 | 9.41 |
| ■ bycyclist (0.03%) | 2.08 | 4.91 | 2.90 | 4.56 | 6.94 | 3.59 |
| ■ road (29.62%) | 64.75 | 53.21 | 43.02 | 67.94 | 58.06 | 47.07 |
| ■ sidewalk (6.65%) | 31.96 | 28.02 | 22.31 | 40.72 | 35.88 | 26.53 |
| ■ other-grnd(11.40%) | 22.16 | 21.65 | 19.78 | 27.02 | 27.83 | 23.67 |
| ■ building (18.38%) | 10.30 | 14.75 | 13.53 | 18.06 | 21.20 | 17.91 |
| ■ lane-marking (1.15%) | 20.47 | 18.10 | 14.78 | 22.97 | 21.65 | 16.73 |
| ■ vegetation (22.41%) | 14.28 | 13.24 | 12.75 | 24.02 | 22.39 | 18.28 |
| ■ trunk (0.90%) | 2.69 | 3.31 | 2.53 | 10.51 | 8.20 | 4.84 |
| ■ pole (0.72%) | 1.77 | 3.18 | 2.96 | 12.09 | 9.48 | 5.98 |
| ■ traf.-sign (0.19%) | 1.71 | 2.32 | 2.53 | 7.29 | 5.71 | 4.25 |
| ■ other-obejct (0.03%) | 4.99 | 7.70 | 4.65 | 9.97 | 6.93 | 4.35 |

(see the 3rd row). However, when the FOV is limited, MonoScene fails to predict the scene outside the visible input (the last two rows). Additionally, in order to qualitatively analyze results under different camera settings, We show the comparison of semantic scene completion results between monocular and trinocular methods on our proposed SSCBench-Waymo dataset, as Fig. II, III, IV and V. Benefiting from the more visual input of the supplementary perspectives, trinocular method can also have a better perception of the environment next to the ego vehicle. Moreover, multi-view input also contributes to more reasonable reasoning for the front view, especially the details in long-range, as shown in the second column of Fig. II.

Table V: **Cross-domain evaluation of LMSCNet (Roldao et al., 2020)**. We report the experiment results of training and testing on different datasets using unified labels. Each dataset name is color-highlighted for clearer reading.

| Training dataset | SSCBench-KITTI-360 | | | | | | SSCBench-Waymo | | | | | |
|---|---|---|---|---|---|---|---|---|---|---|---|---|
| Testing dataset | SSCBench-KITTI-360 | | | SSCBench-Waymo | | | SSCBench-KITTI-360 | | | SSCBench-Waymo | | |
| Range | 12.8m | 25.6m | 51.2m | 12.8m | 25.6m | 51.2m | 12.8m | 25.6m | 51.2m | 12.8m | 25.6m | 51.2m |
| IoU (%) | 67.44 | 58.85 | 48.49 | 16.06 | 16.28 | 17.45 | 8.44 | 5.89 | 4.12 | 89.80 | 89.80 | 89.59 |
| Precision (%) | 75.60 | 74.90 | 74.82 | 19.86 | 20.22 | 21.81 | 22.99 | 25.62 | 27.24 | 95.73 | 95.73 | 95.63 |
| Recall (%) | 81.53 | 73.31 | 57.95 | 45.62 | 45.54 | 46.63 | 11.76 | 7.10 | 4.62 | 93.55 | 93.54 | 93.42 |
| mIoU | 32.84 | 28.69 | 22.48 | 1.71 | 1.75 | 1.83 | 2.41 | 1.59 | 1.04 | 52.42 | 52.41 | 51.98 |
| ▪ vehicle | 41.87 | 35.37 | 23.87 | 0.37 | 0.41 | 0.36 | 3.82 | 2.35 | 1.34 | 75.62 | 75.56 | 76.56 |
| ▪ bicycle | 0.00 | 0.00 | 0.00 | 0.00 | 0.00 | 0.00 | 0.00 | 0.00 | 0.00 | 0.00 | 0.00 | 0.00 |
| ▪ motorcycle | 0.00 | 0.00 | 0.00 | 0.00 | 0.00 | 0.00 | 0.00 | 0.00 | 0.00 | 0.00 | 0.00 | 0.00 |
| ▪ person | 0.00 | 0.00 | 0.00 | 0.00 | 0.00 | 0.00 | 0.07 | 0.03 | 0.03 | 65.23 | 64.23 | 62.69 |
| ▪ road | 78.45 | 75.47 | 64.04 | 4.07 | 4.19 | 4.02 | 3.15 | 2.34 | 1.65 | 79.70 | 80.33 | 79.96 |
| ▪ sidewalk | 53.53 | 46.61 | 35.13 | 0.84 | 0.86 | 0.82 | 0.75 | 0.36 | 0.19 | 52.42 | 52.29 | 52.37 |
| ▪ other-grnd | 30.05 | 25.41 | 17.46 | 2.31 | 2.36 | 2.77 | 1.81 | 1.34 | 0.90 | 52.37 | 52.44 | 52.27 |
| ▪ building | 60.71 | 54.30 | 44.47 | 2.31 | 2.36 | 2.74 | 6.42 | 5.17 | 4.03 | 72.97 | 72.95 | 72.45 |
| ▪ vegetation | 58.65 | 49.69 | 39.75 | 9.19 | 9.35 | 9.99 | 8.05 | 4.22 | 2.19 | 75.47 | 75.28 | 74.94 |
| ▪ other-obejct | 0.16 | 0.07 | 0.04 | 0.01 | 0.00 | 0.00 | 0.06 | 0.10 | 0.06 | 50.49 | 50.02 | 48.54 |

Table VI: **Cross-domain evaluation of SSCNet (Song et al., 2017)**.

| Training dataset | SSCBench-KITTI-360 | | | | | | SSCBench-Waymo | | | | | |
|---|---|---|---|---|---|---|---|---|---|---|---|---|
| Testing dataset | SSCBench-KITTI-360 | | | SSCBench-Waymo | | | SSCBench-KITTI-360 | | | SSCBench-Waymo | | |
| Range | 12.8m | 25.6m | 51.2m | 12.8m | 25.6m | 51.2m | 12.8m | 25.6m | 51.2m | 12.8m | 25.6m | 51.2m |
| IoU (%) | 70.98 | 63.56 | 54.50 | 16.79 | 16.99 | 17.82 | 10.52 | 9.01 | 8.99 | 85.83 | 85.86 | 85.89 |
| Precision (%) | 76.88 | 72.21 | 70.07 | 22.83 | 23.20 | 24.58 | 25.01 | 27.19 | 30.43 | 89.26 | 89.30 | 89.31 |
| Recall (%) | 90.25 | 84.13 | 71.03 | 38.85 | 38.82 | 39.31 | 15.37 | 11.87 | 11.31 | 95.70 | 95.71 | 95.72 |
| mIoU | 35.03 | 31.68 | 25.63 | 4.18 | 4.20 | 4.18 | 3.00 | 2.07 | 1.33 | 51.59 | 51.51 | 51.31 |
| ▪ vehicle | 49.54 | 44.31 | 34.14 | 3.92 | 3.69 | 3.11 | 7.00 | 4.62 | 2.73 | 69.80 | 71.00 | 71.55 |
| ▪ bicycle | 0.00 | 0.00 | 0.00 | 0.00 | 0.00 | 0.00 | 0.00 | 0.00 | 0.00 | 6.57 | 5.43 | 5.65 |
| ▪ motorcycle | 0.00 | 0.00 | 0.00 | 0.00 | 0.00 | 0.00 | 0.00 | 0.00 | 0.00 | 0.00 | 0.00 | 0.00 |
| ▪ person | 0.43 | 0.19 | 0.11 | 0.00 | 0.00 | 0.00 | 0.09 | 0.05 | 0.04 | 61.96 | 60.94 | 60.26 |
| ▪ road | 79.71 | 77.76 | 67.54 | 4.24 | 4.12 | 3.47 | 4.05 | 2.67 | 1.73 | 78.37 | 78.97 | 78.65 |
| ▪ sidewalk | 58.95 | 53.18 | 41.84 | 1.62 | 1.65 | 1.47 | 0.94 | 0.49 | 0.27 | 53.42 | 53.10 | 53.11 |
| ▪ other-grnd | 33.70 | 28.96 | 19.86 | 2.39 | 2.37 | 2.31 | 1.32 | 0.75 | 0.50 | 52.88 | 53.17 | 53.02 |
| ▪ building | 64.49 | 57.39 | 46.89 | 16.96 | 17.42 | 18.20 | 7.11 | 7.16 | 5.53 | 70.53 | 70.47 | 70.17 |
| ▪ vegetation | 61.66 | 53.99 | 44.92 | 12.10 | 12.20 | 12.77 | 9.39 | 4.82 | 2.46 | 72.13 | 72.03 | 71.99 |
| ▪ other-obejct | 1.79 | 1.66 | 1.01 | 0.59 | 0.55 | 0.44 | 0.05 | 0.10 | 0.07 | 50.22 | 49.95 | 48.73 |

Table VII: **Cross-domain evaluation of MonoScene (Cao & de Charette, 2022)**.

| Training dataset | SSCBench-KITTI-360 | | | | | | SSCBench-Waymo | | | | | |
|---|---|---|---|---|---|---|---|---|---|---|---|---|
| Testing dataset | SSCBench-KITTI-360 | | | SSCBench-Waymo | | | SSCBench-KITTI-360 | | | SSCBench-Waymo | | |
| Range | 12.8m | 25.6m | 51.2m | 12.8m | 25.6m | 51.2m | 12.8m | 25.6m | 51.2m | 12.8m | 25.6m | 51.2m |
| IoU (%) | 57.39 | 46.81 | 39.11 | 11.68 | 9.7 | 7.07 | 16.03 | 13.41 | 14.72 | 63.37 | 46.1 | 54.61 |
| Precision (%) | 70.86 | 64.27 | 60.84 | 19.13 | 15.13 | 11.95 | 28.70 | 25.18 | 25.20 | 80.18 | 64.18 | 53.46 |
| Recall (%) | 75.11 | 63.28 | 52.26 | 23.07 | 21.38 | 14.74 | 26.64 | 22.30 | 26.14 | 75.14 | 62.07 | 54.61 |
| mIoU | 32.01 | 25.43 | 19.53 | 2.84 | 2.78 | 2.16 | 2.56 | 2.47 | 2.28 | 18.17 | 17.37 | 14.67 |
| ▪ vehicle | 35.27 | 29.50 | 21.29 | 6.07 | 8.6 | 9.87 | 1.24 | 1.18 | 0.98 | 20.59 | 20.08 | 16.09 |
| ▪ bicycle | 7.59 | 3.89 | 2.60 | 0.54 | 0.15 | 0.01 | 0.00 | 0.00 | 0.00 | 3.83 | 3.29 | 2.03 |
| ▪ motorcycle | 7.01 | 3.02 | 1.60 | 0.00 | 0.00 | 0.00 | 0.00 | 0.00 | 0.00 | 0.00 | 0.17 | 0.09 |
| ▪ person | 6.36 | 4.54 | 3.27 | 1.23 | 1.76 | 2.12 | 0.00 | 0.00 | 0.00 | 14.28 | 9.86 | 6.82 |
| ▪ road | 73.02 | 64.39 | 51.62 | 3.44 | 4.02 | 2.99 | 13.14 | 8.48 | 6.58 | 69.66 | 57.12 | 46.31 |
| ▪ sidewalk | 48.87 | 39.37 | 30.42 | 1.14 | 0.95 | 0.55 | 3.80 | 2.51 | 0.19 | 32.46 | 28.31 | 23.14 |
| ▪ other-grnd | 30.39 | 21.08 | 14.25 | 0.56 | 0.65 | 0.24 | 1.94 | 2.85 | 2.37 | 19.27 | 22.87 | 20.92 |
| ▪ building | 51.44 | 43.12 | 34.83 | 5.78 | 3.28 | 0.45 | 0.76 | 4.23 | 5.11 | 5.21 | 14.23 | 13.7 |
| ▪ vegetation | 48.24 | 36.48 | 29.16 | 8.92 | 7.70 | 5.00 | 2.82 | 4.20 | 5.26 | 13.66 | 13.89 | 13.9 |
| ▪ other-obejct | 11.95 | 8.88 | 6.26 | 0.74 | 0.63 | 0.38 | 0.01 | 0.01 | 0.02 | 2.73 | 3.86 | 3.65 |

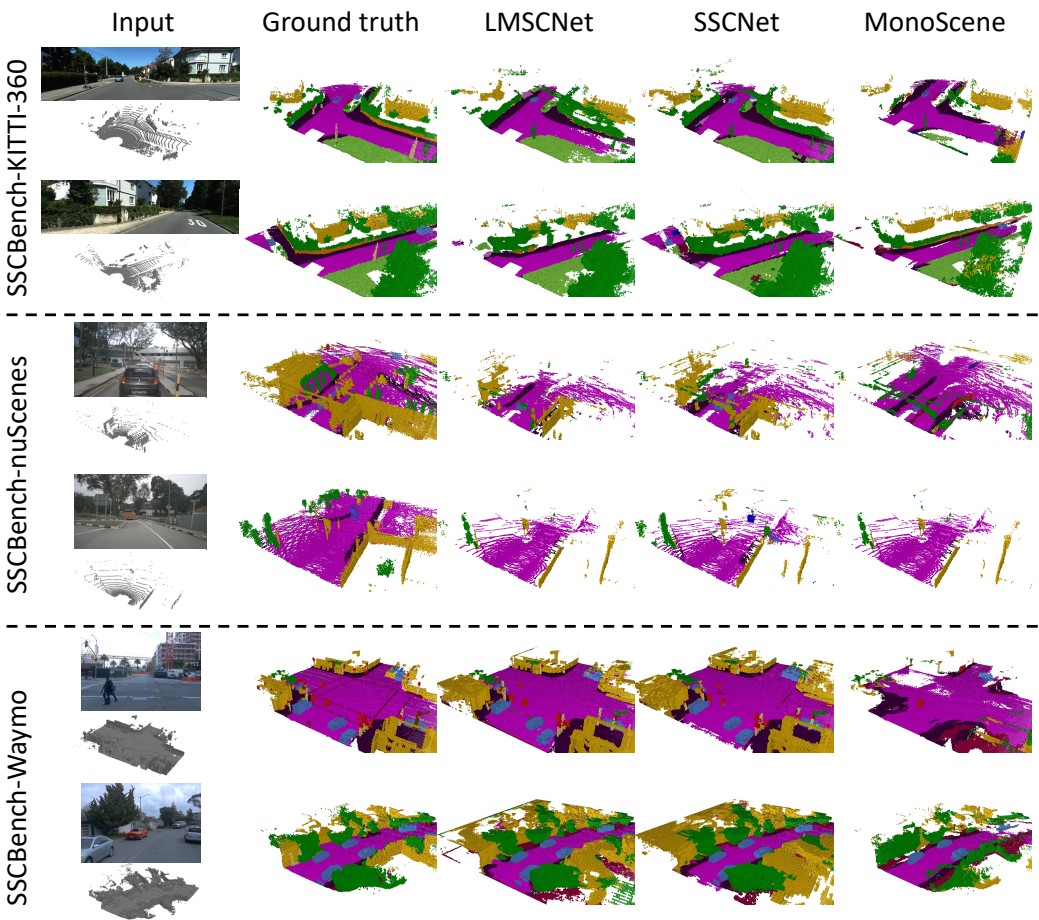

Figure I: **Qualitative experiment results.** We show the semantic scene completion results of different methods on our proposed SSCBench dataset. The predictions of each method are compared with the input and the ground truth.

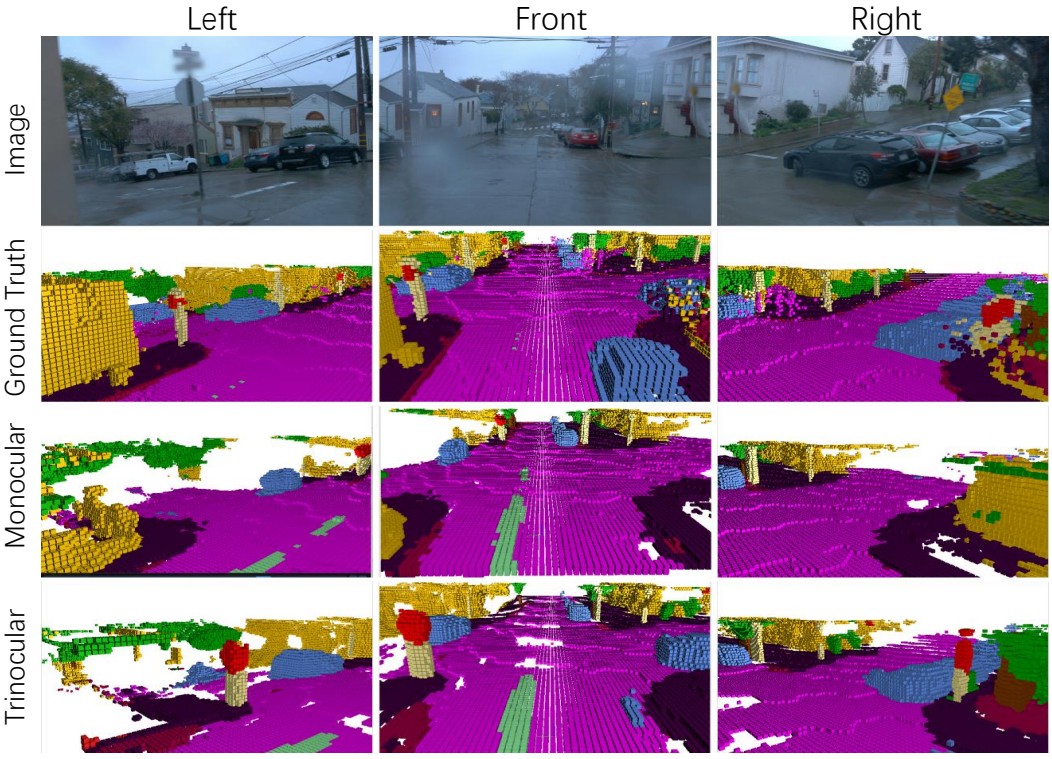

Figure II: **Qualitative comparison between monocular and trinocular setup.** We show the semantic scene completion results of TPVFormer with different setups on SSCBench-Waymo.

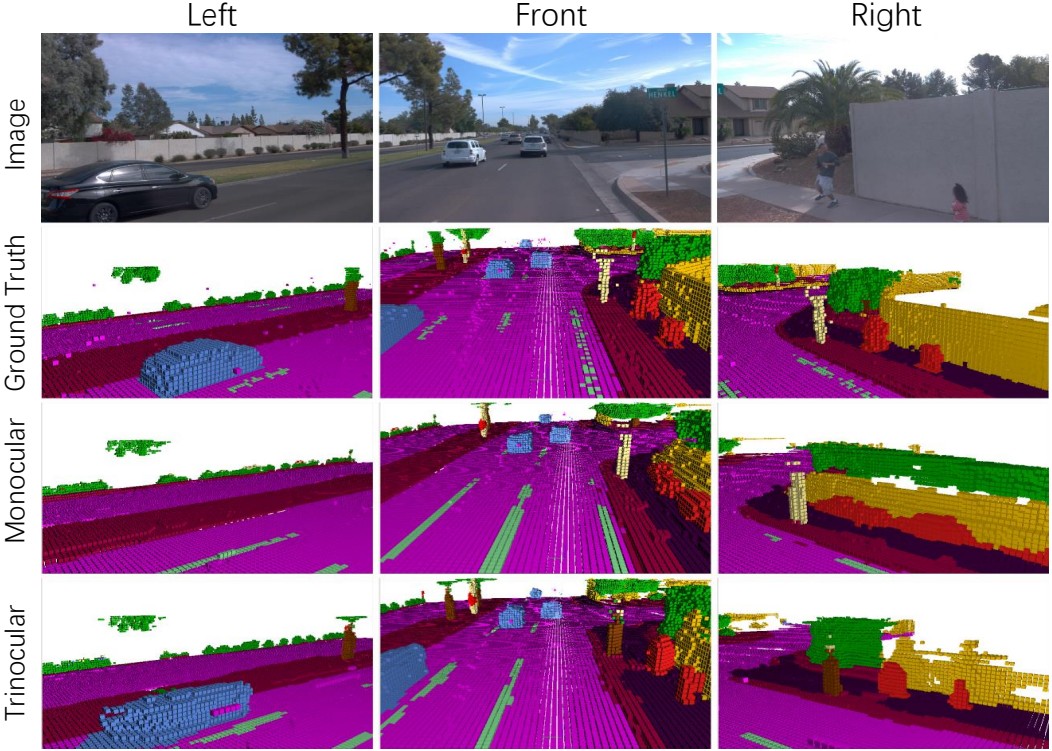

Figure III: **Qualitative comparison between monocular and trinocular setup.** We show the semantic scene completion results of TPVFormer with different setups on SSCBench-Waymo.

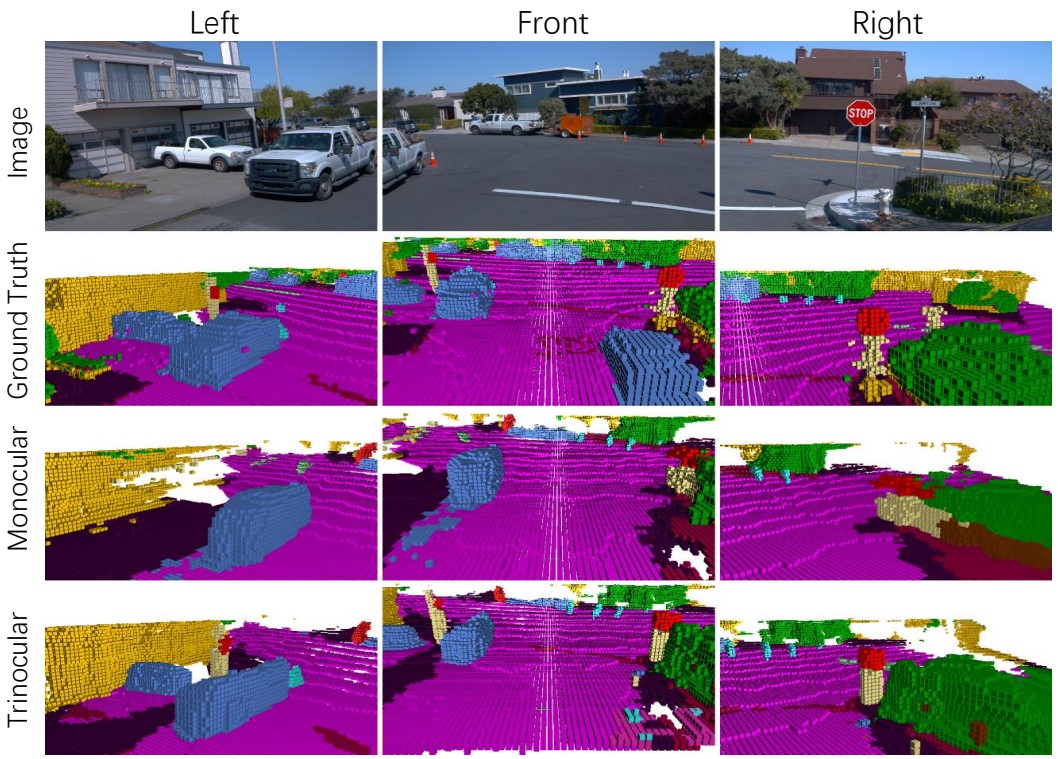

Figure IV: **Qualitative comparison between monocular and trinocular setup.** We show the semantic scene completion results of TPVFormer with different setups on SSCBench-Waymo.

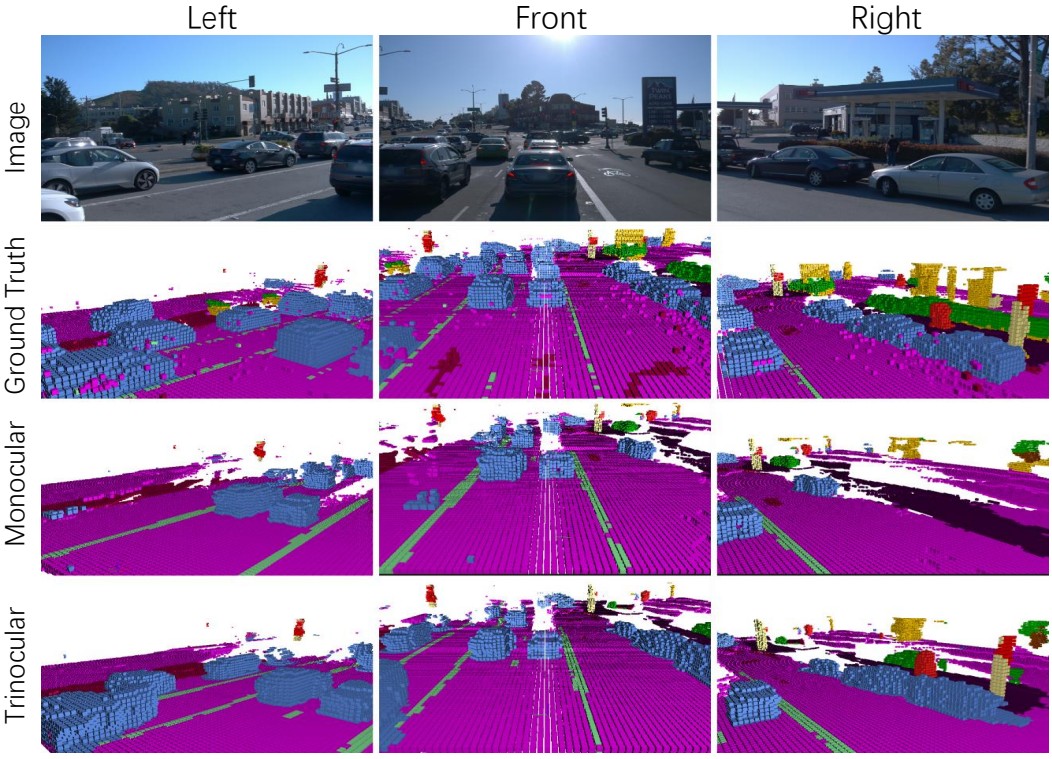

Figure V: **Qualitative comparison between monocular and trinocular setup.** We show the semantic scene completion results of TPVFormer with different setups on our proposed SSCBench-Waymo.

