# OpenReview forum: "SSCBench: Monocular 3D Semantic Scene Completion Benchmark in Street Views"
_ICLR.cc/2024/Conference — ICLR 2024 Conference Withdrawn Submission_

### Official Review · Reviewer_JQ2D · 2023-10-26

**Soundness:** 3 good
**Presentation:** 3 good
**Contribution:** 2 fair
**Rating:** 5
**Confidence:** 4

**Summary:**

This paper introduces SSCBench, a novel benchmark for the semantic scene completion task. SSCBench integrates scenes from three prominent automotive datasets: KITTI-360, nuScenes, and Waymo. The benchmark evaluates models using monocular, trinocular, and point cloud inputs, providing a comprehensive analysis of sensor coverage and modality.

**Strengths:**

The paper is clearly written and conducts extensive experiments on three SSC benchmarks. The large scale benchmark contributes to the research community's study of dense semantic prediction in street views. The study of cross-domain evaluation is meaningful and can facilitate future unified 3D models.

**Weaknesses:**

Concerns on the novelty of the benchmark:

Based on the benchmark construction pipeline, it appears that SSCBench does not exhibit a significant deviation from recent occupancy datasets, like OpenOccupancy, SurroundOcc, and Occ3D. These methodologies, including the one proposed in this paper, rely on the temporal aggregation of Lidar points and incorporate foreground boxes for dynamic object processing. In contrast, SurroundOcc employs mesh, while OpenOccupancy uses manual annotation to enhance voxel quality. While the authors elucidate the differences in a paragraph, the process of transitioning from a 6-views setting to monocular, trinocular, or LiDAR settings seems relatively straightforward. In terms of scale, this benchmark merely adds the KITTI dataset in comparison to Occ3D. Compared to evaluating each dataset separately, building an integrated benchmark sounds more meaningful.

Concerns on the methods in the benchmark:

Given that completion is more challenging and requires inferring occluded regions in the current frame, a temporal method becomes necessary. The approach of predicting invisible structures from a single frame seems not very reasonable.

Concerns on the application of the benchmark:

This is an open question. What are the benefits of dense semantic prediction in the field of autonomous driving? Is sparse semantic perception sufficient? Because the performance in this paper shown that 3D semantics is very poor (low mIoU). What are the benefits of this type of benchmark for future research in this field.

**Questions:**

1. Unknown voxels depicted in Figure2 may be slightly ambiguous. I guess it implies that the occluded road and all empty voxels  are regarded as “Unknown voxels ”?
2. How to handle the hollow part in the middle of an object, as lidar only hits the surface? For example, a bus maybe only can hit one surface, how to deal with such cases.

---

### Official Review · Reviewer_5Bcg · 2023-10-30

**Soundness:** 3 good
**Presentation:** 4 excellent
**Contribution:** 3 good
**Rating:** 8
**Confidence:** 3

**Summary:**

This paper introduces SSCBench, a novel benchmark for monocular semantic scene completion (SSC). The benchmark takes data from KITTI-360, nuScenes, and Waymo to build up three large-scale datasets. Designed to be compatible with SemanticKITTI, SSCBench addresses two key issues inherent in the latter: the accuracy of ground truth values impacted by dynamic objects and a limited diversity of urban scenes. The SSCBench datasets are 7.7 times larger than SemanticKITTI and incorporate advancements such as dynamic object synchronization and unknown voxel exclusion. The paper also evaluates four existing SSC methods using these newly generated datasets.

**Strengths:**

+ The two techniques proposed in the paper, dynamic object synchronization and unknown voxel exclusion, ensure the dataset's quality. The curated dataset incorporates three open-source datasets and covers various geolocations.
+ The authors provide a comprehensive benchmark for different SSC methods, input modalities, and cross-domain adaptation performance.
+ The proposed dataset is in the format of SemanticKITTI, which is a classic dataset for SSC. This makes benchmarking and evaluation of the dataset accessible to existing and future SSC methods.

**Weaknesses:**

- As mentioned in the limitation, the current dataset only utilized 3D data, limiting the development and evaluation of 4D methods.

**Questions:**

- The current benchmark only focuses on the performance of different SSC methods. It would be meaningful to also benchmark their computational requirements in terms of training time, memory requirements, FLOPs, etc.
- The cross-domain benchmark only considers training and testing on different datasets. However, as the authors mentioned they unified the labels across the dataset, so it would be intuitive to train on multiple datasets and report the corresponding performance. Otherwise, the claimed diversity of the dataset is not supported by the benchmark.

---

### Official Review · Reviewer_MTom · 2023-10-30

**Soundness:** 3 good
**Presentation:** 3 good
**Contribution:** 2 fair
**Rating:** 5
**Confidence:** 4

**Summary:**

This paper presents a monocular 3D semantic scene completion benchmark in the focus of autonomous driving scenarios. The benchmark covers three prevailing datasets, which are KITTI-360, nuScenes and Waymo. Then the author benchmarks models from the input of single-front-camera, three-front-cameras and the LiDAR. Meanwhile, the author also tries to unify labels across different dataset, though this part remains concern.

**Strengths:**

1. The work involves many datasts, which requires substantial workload. And the author commits to including more datasets and SSC models to drive further advancements in this field, which makes sense to the community.
2. The work benchmarks 6 models from different input type, including single-front-camera, three-front-cameras and the LiDAR. This also requires substantial workload.
3. The paper writing is clear and easy to understand.

**Weaknesses:**

1. Not big difference to its related work. The only biggest difference between this work and Occ3D / OpenOccupancy is that this work limits the input to monocular style, which is pretty easy to be done on those other work (just simply mask out those voxels in other views).
2. The motivation of predicting occluded area from camera input. This is rather not intuitive. What’s the motivation of doing this? Does the downstream task need it or is it just more challenging but meaningless?
3. Missing some related work. Some occupancy work like FB-OCC[1], OccDepth[2], OccNet[3], PanoOcc[4] and so on.
4. Missing detail for nuscenes aggregation. How do the author do with the missing z-axis in the meta-data and thus compromising the data quality.
5. What’s the point of comparing monocular methods and trinocular methods? Are they on the same test set? Are they using the same network structure?
6. Why on the ssc-benchmark, the performance of those methods are different from those on semantic-kitti? This is not the same as what’s discovered in those occupancy work such as Occ3D and OpenOccupancy.
7. Unifying different datasets are meaningful.but how is the label / range unified? It would be very helpful to have more detail here.

[1]: FB-OCC: 3D Occupancy Prediction based on Forward-Backward View Transformation. Zhiqi Li, Zhiding Yu, David Austin, Mingsheng Fang, Shiyi Lan, Jan Kautz, Jose M. Alvarez
[2]: OccDepth: A Depth-Aware Method for 3D Semantic Scene Completion. Ruihang Miao, Weizhou Liu, Mingrui Chen, Zheng Gong, Weixin Xu, Chen Hu, Shuchang Zhou.
[3]: Scene as Occupancy. Wenwen Tong, Chonghao Sima, Tai Wang, Silei Wu, Hanming Deng, Li Chen, Yi Gu, Lewei Lu, Ping Luo, Dahua Lin, Hongyang Li
[4]: PanoOcc: Unified Occupancy Representation for Camera-based 3D Panoptic Segmentation. Yuqi Wang, Yuntao Chen, Xingyu Liao, Lue Fan, Zhaoxiang Zhang.

**Questions:**

1. What’s the point of comparing monocular methods and trinocular methods? Are they on the same test set? Are they using the same network structure?
2. Why on the ssc-benchmark, the performance of those methods are different from those on semantic-kitti? This is not the same as what’s discovered in those occupancy work such as Occ3D and OpenOccupancy.
3. Unifying different datasets are meaningful.but how is the label / range unified? It would be very helpful to have more detail here.

---

### Official Review · Reviewer_drPb · 2023-11-06

**Soundness:** 3 good
**Presentation:** 2 fair
**Contribution:** 3 good
**Rating:** 5
**Confidence:** 4

**Summary:**

The paper proposes a benchmark to enable unified SSC evaluation from three existing street-view datasets. This is constructed using a pipeline of aggregation, voxelisation and post-processing. Evaluation using latest SSC methods is included.

**Strengths:**

* Targets active research field of AD, where more useful data and insights are appreciated by the community, and unification is desirable
* Well structured and readable
* Includes latest SOTA methods for comparison
* Extensive comparisons and analysis of results
* Dynamic labels like person seem well handled

**Weaknesses:**

* Sec. 3.3 lacks enough detail to reproduce the paper, i.e. synchronization and perspective probing.
* GT exhibits unsatisfactory level of noise, e.g. floating road voxels in Fig. V. There should be consistent and universally applied occupancy filters (i.e. class-specific rule-based) to fix such obvious cases as a part of the unification process.
* Similar for semantic label consistency - lane label GT continuity (e.g. Fog V) is lost as part of the voxelisation process and should be avoided using smarter voting scheme to preserve linear structures (also poles).
* Actually, methods themselves (i.e. TPVFormer) already do a better job at the above, and you penalize them for doing so, which is not acceptable (especially when avoidable)
* While cross-domain results are insightful, they generate an explosion of results, which are difficult to refer to. A unified benchmark should also produce a single unified metric, and associated leaderboard as a result. In this case, I'd suggest something like five-fold cross-validation over random but fixed splits sampled uniformly across all included datasets.
* Figs. II-V voxel renders are from a different viewpoint/FOV than Image row, which is very confusing, e.g. Fig II centre there is a big car near the camera in GT but nothing in image.
* This is a vision dataset paper, which does not contribute to representation methodology and as such of limited significance to a wider audience

**Questions:**

* Do you plan to host online evaluation and leaderboards?